# Genomic epidemiology offers high resolution estimates of serial intervals for COVID-19

Jessica E. Stockdale [1] ✉, Kurnia Susvitasari [1], Paul Tupper[1], Benjamin Sobkowiak [1], Nicola Mulberry[1], Anders Gonçalves da Silva[2,4], Anne E. Watt [2], Norelle L. Sherry [2], Corinna Minko[3], Benjamin P. Howden [2], Courtney R. Lane[2,5] & Caroline Colijn[1,5]

Serial intervals – the time between symptom onset in infector and infectee – are a fundamental quantity in infectious disease control. However, their estimation requires knowledge of individuals' exposures, typically obtained through resource-intensive contact tracing efforts. We introduce an alternate framework using virus sequences to inform who infected whom and thereby estimate serial intervals. We apply our technique to SARS-CoV-2 sequences from case clusters in the first two COVID-19 waves in Victoria, Australia. We find that our approach offers high resolution, cluster-specific serial interval estimates that are comparable with those obtained from contact data, despite requiring no knowledge of who infected whom and relying on incompletely-sampled data. Compared to a published serial interval, cluster-specific serial intervals can vary estimates of the effective reproduction number by a factor of 2–3. We find that serial interval estimates in settings such as schools and meat processing/packing plants are shorter than those in healthcare facilities.

Whole-genome sequence (WGS) data is rapidly becoming a fundamental tool in public health laboratories (PHL) around the world[1–3]. WGS data carry enormous benefits for outbreak investigations: identifying transmission events that were not detected during epidemiological study[4] and revealing the impact of border control measures[5], especially where data are shared across jurisdictional boundaries[3]. However, the information content of genomic data alone can be limited, as experienced during the SARS-CoV-2 pandemic[6,7]. Often, genomic data are combined with epidemiological data in an ad hoc fashion by plotting epidemiological data on the tips of phylogenetic trees (derived from genomic data). This does not lend itself readily to desired PHL reproducibility and repeatability standards. On the other hand, when genomic surveillance data are systematically linked to epidemiological and clinical information, genomic epidemiological investigations can better inform public health action through a contextual understanding of population demographics, immunisation, clinical impacts, spatial transmission patterns, and more[8,9].

In this work, we develop a framework for the integration of genomic data, in the form of whole-genome virus sequences, into epidemiological investigations, particularly when detailed epidemiological data from contact tracing is unavailable. Pathogen sequence data collected from infected individuals do not directly reveal who infected whom, but nonetheless can offer a high-resolution view of transmission. We focus on cluster-specific estimation of the serial interval, a key measure describing the spread of an infectious disease, which is defined as the length of time between the onset of symptoms in a primary and secondary case. This is informative of both the speed

[1]Department of Mathematics, Simon Fraser University, Burnaby, BC, Canada. [2]Microbiological Diagnostic Unit Public Health Laboratory, Department of Microbiology & Immunology, University of Melbourne at the Peter Doherty Institute for Infection & Immunity, Melbourne, VIC, Australia. [3]Victorian Department of Health, Melbourne, VIC, Australia. [4]Deceased: Anders Gonçalves da Silva. [5]These authors jointly supervised this work: Courtney R. Lane, Caroline Colijn. ✉e-mail: jessica_stockdale@sfu.ca

of transmission, as well as when in the infection process transmission is likely to occur.

Serial intervals are typically inferred from small clusters of individuals with known contact and times of symptom onset[10,11], but the collection of such data can be resource intensive, and privacy and reporting considerations limit wide reporting and utilisation. As a result, estimates of the serial interval applied in practice (underpinning estimation of other epidemiological quantities such as the time-dependent reproduction number $R_t$) are often taken from small studies, not necessarily from the same location or time as the population in question. Methods that do not require knowledge of who infected whom have been developed, but these assume that the population is fully sampled[12,13]. When contact tracing data is available, common approaches are to consider the distribution of observed serial intervals between contact-traced pairs assumed to represent direct transmission[14], or to monitor the population for index cases who are infected with the pathogen of interest, and then follow up with close contacts such as members of their household to find secondary cases[15]. Such approaches were extended by ref. 10 to allow for unsampled intermediate cases, using index case-to-case (ICC) intervals, defined as the lengths of time between symptom onset of all secondary cases and the index case in a small population such as a household, boarding school or closed workplace. By allowing for up to two unsampled cases between the index case and a secondary case, Vink et al. take potential under-reporting into account. However, the limitation on the number of unsampled intermediates and the onus on the identification of the index case mean that this approach is most suited to small and closed populations.

We present a framework that uses virus sequences in place of direct knowledge of infection pairs, for inference of the serial interval distribution in incompletely-sampled case clusters. Our approach does not restrict the number of unsampled intermediate cases, or make assumptions about the infectious period or latent distribution. It allows for the possibility of presymptomatic transmission but does assume serial intervals are positive. We incorporate uncertainty in who infected whom by first sampling a set of feasible transmission networks given the virus sequences and the known times of symptom onset, but we require no knowledge of contact between individuals in the clusters. We then use a mixture model for estimation of the serial interval, which takes into account that the outbreak may not be fully sampled and so inferred transmission pairs may not represent direct transmission. While there exist several algorithms for outbreak reconstruction from genomic data in the context of sufficient genetic variation to construct well-resolved pairs or phylogenetic trees, for example, the outbreaker, TransPhylo, SCOTTI and Beastlier platforms[16–19], these have been focused on densely-sampled outbreak settings where inference of who infected whom is the primary aim. Our approach is targeted at broader settings with lower levels of sampling and genetic variation, where there may not be sufficient information in the data to reconstruct transmission pathways with high confidence. We, therefore, use a fast and simple model for pair reconstruction, followed by a statistical model that averages over the uncertainty in who infected whom to estimate the serial interval distribution. To the best of our knowledge, no existing outbreak reconstructive models have considered the estimation of serial intervals in this context. We demonstrate that virus sequences offer a practical approach for inference of cluster-specific estimates, although our methods are also appropriate in broader settings, even where detailed contact tracing data are not available and whilst taking under-reporting into account.

We investigate the use of virus sequences for the estimation of serial intervals using SARS-CoV-2 whole-genome sequences and recorded symptom onset times from Victoria, Australia. We identify a number of genomically- and epidemiologically-defined SARS-CoV-2 clusters from the first and second waves of the COVID-19 pandemic in Victoria: with samples collected from 6 January–14 April 2020 and 1 June–28 October 2020. We estimate the serial interval in each cluster, allowing for comparison both within and between waves. We additionally compare estimates of the serial interval arising from different types of cluster, including healthcare facilities and workplaces. We explore the impact that using cluster-specific serial interval estimates has on downstream estimates of the time-dependent reproduction number $R_t$. We compare our sequence-based approach with an equivalent method that uses the same model but detailed contact-tracing information in place of virus sequences, and we validate the method against simulated outbreaks where the proportion of missing cases can be controlled.

## Results

### Estimation of the serial interval using pathogen sequences

We introduce a new framework for serial interval estimation, using pathogen sequences from a transmission cluster of interest to infer who infected whom, whilst taking uncertainty and incomplete sampling into account. A schematic diagram of the method is shown in Fig. 1, and further details are provided in Methods. Whereas existing approaches use contact data to infer the transmission tree, we use a set of pathogen sequences along with cases' symptom onset times to sample a set of

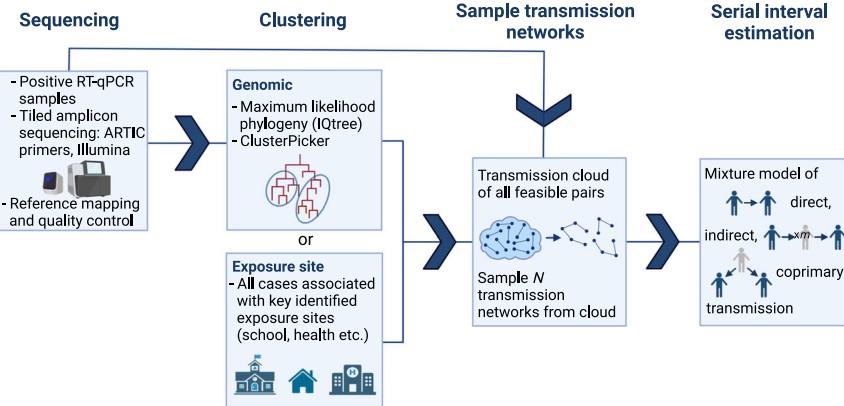

**Fig. 1 | Model schematic.** Description of the overall methodology pipeline. After sequencing, cases are clustered either genomically or epidemiologically. The serial interval is estimated in each cluster, by repeatedly sampling plausible transmission networks from among a set of plausible transmission pairs, estimating the serial interval in each network and combining the estimates. The serial interval estimation takes into account that each cluster is not fully sampled. A full description of each stage is provided in Methods.

plausible transmission networks, accounting for indirect transmission, and thereby estimate the serial interval. This is intended to be applied at the level of relatively broad clusters, where we see sustained transmission but are not necessarily able to sample every case.

After sequencing and clustering our cases, we first create a 'transmission cloud' per cluster. That is, a set of all plausible transmission pairs (infectee/infector pairs) in the cluster, who meet predetermined criteria for genomic distance and distance between symptom onset times. From the cloud, we obtain a set of plausible transmission networks by repeatedly sampling an infector for each infectee with probability inversely proportional to their genomic and symptom onset time distance. We take incomplete sampling into account by fitting a mixture model for the serial interval to each sampled transmission network; this incorporates that when $i$ is the sampled infector of $j$, transmission from $i$ to $j$ may have been direct, indirect through an unknown number of unsampled intermediate cases, or $i$ and $j$ may have both been infected by the same unsampled individual (coprimary infection). We combine the serial interval estimates from the mixture model across all sampled networks in the cluster, to obtain a final cluster-specific estimate of the serial interval distribution. Overall, we assume that the serial interval follows a Gamma distribution with mean $\mu$ and standard deviation $\sigma$. We further assume that sampled infector $i$ and infectee $j$ are separated by some $m$ unsampled individuals, $0 \leq m < \infty$, where $m$ follows a geometric distribution with parameter $\pi$. Then, $\pi$ can be thought of as a sampling probability within an identified transmission chain. A proportion $w$ of transmission pairs are of this type, with the remaining proportion $(1 - w)$ being of coprimary type: $i$ and $j$ were infected by the same unsampled individual. Parameter $w$ is therefore also related to the cluster case ascertainment rate, in that we failed to observe $i$ and $j$'s mutual infector. We use maximum a posteriori estimation (MAP) to obtain estimates of the parameters ($\mu$, $\sigma$, $\pi$, $w$) from the mixture model, with prior distributions on $\pi$ and $w$ to incorporate our knowledge on the rate of population sampling and thereby constrain the estimation procedure.

All serial interval analysis is performed using R version 4.1.0. The code is available at github.com/jessicastockdale/genomicSIs[20].

## Validation against simulated data

Before estimating the serial interval in real data, we validate our approach using simulated outbreaks with known serial interval distribution. We simulate an influenza-like outbreak with $R_0 = 2$ and serial interval ~$\Gamma(\mu = 4.5, \sigma = 2)$, using the outbreaker package in R[21] to generate symptom onset times and pathogen sequences. We run 10 experiments in which we increasingly down-sample the simulated outbreak, by retaining a proportion $p = 1.0, 0.9, \ldots, 0.1$ of infected cases, to explore the impact of incomplete sampling on our estimates. Full details of the procedure are given in Methods, and further simulation studies in which we explore the ability of our method to distinguish serial intervals with different mean and under different prior distributions are included in the Supplemental Materials.

Results of the simulation study are shown in Fig. 2. In all experiments, we find that our method is able to estimate the mean serial interval well, but with increasing uncertainty as the proportion of cases decreases. The results are similar for the serial interval standard deviation, with some upward bias for low $p$. The sampling proportion $p$ in our experiments is not exactly equivalent to either the true sampling probability $\pi$ or proportion non-coprimary $w$ being estimated in the model, however, they are closely related. This is found in our model estimates, with increasing $\pi$ and $w$ estimates under higher $p$. Again, our estimates have higher uncertainty as the amount of down-sampling increases.

## SARS-CoV-2 whole-genome sequencing and clustering, Victoria, Australia

We estimate the serial interval in transmission clusters from the first (6 January–14 April 2020) and second (1 June–28 October 2020) waves

of the COVID-19 pandemic in Victoria. The data comprise genomic sequence, sequence sampling date and symptom onset date for each sampled case. See Methods for full details on the sequencing and clustering procedure. Sequences are clustered genomically in the first wave and epidemiologically in the second wave: the epidemiological clustering procedure does not use any detailed demographic or contact data; we group cases associated with exposure sites defined by public health, including schools, healthcare and workplaces. The genomic clustering procedure in wave 1 was found to have strong concordance with an epidemiological clustering approach for this data[6], with a median of 100% of cases in each epidemiologically-linked group being in a single genomic cluster.

A total 1242 samples from 1075 patients were sequenced during the wave 1 study period. This corresponds to 80.7% of identified COVID-19 cases in Victoria during that time period. Of the 903 samples passing quality control and de-duplication, 312 were identified as belonging to a genomic cluster with at least 15 cases, for a total of ten wave 1 clusters. A 15,665 samples from 14,075 patients were sequenced during the wave 2 study period, corresponding to 83.9% of identified Victorian COVID-19 cases. Of the 5745 cases passing quality control and de-duplication and with association to at least one exposure site, 3875 were identified as belonging to a cluster with at least 15 cases. There are a total of 94 wave 2 clusters, although, for our main analysis, we focus on ten primary exposure site clusters comprising the largest two clusters associated with each of aged care facilities, healthcare facilities, housing, schools, and meat packing/meat processing plants. Note that wave 2 cases may be associated with more than one exposure site, in which case they will be included in more than one cluster.

Phylogenetic trees, produced from the entire set of sequences in each wave using a custom workflow for building fast SARS-CoV-2 trees available at github.com/MDU-PHL/kovid-trees-nf, are shown in Fig. 3. Whilst some epidemiological wave 2 clusters align well with clades on the tree, we note the low variation in many sequences close to the root and that many clusters span the entire tree. This is indicative of the difficulty in genomically clustering these cases, but also highlights the broad utility of methods which can effectively combine genomic and epidemiological data sources. The ten primary clusters in each wave are summarised in Table 1, and symptom onset curves are shown in Fig. 4.

## Serial intervals in Victorian COVID-19 clusters

We apply our methodology to the Victorian SARS-CoV-2 data by estimating the serial interval in each of the ten primary wave 1 and wave 2 clusters shown in Table 1, as well as the remaining 84 wave 2 clusters. We sample 100 transmission networks per cluster, and perform an additional analysis in which we pool the sampled networks in each wave, to obtain aggregated whole-wave serial interval estimates. We assume a Beta(12, 11) distributed prior for parameters $\pi$ and $w$ in all analyses, with a mean of 0.52 and a standard deviation of 0.1. This is informed by the proportion of identified Victorian COVID-19 cases during the study period that were sampled and sequenced with sufficient quality (57%), combined with a prior belief of a high case finding rate, motivated by detailed case follow-up and high source-acquisition in Victoria at the time[22]. Note that $\pi$ represents the within-transmission-chain sampling probability, rather than the overall sampling probability in the cluster or population at large: it only concerns cases that were not identified or sequenced but did infect others within the cluster. We did not assume any prior knowledge of the serial interval distribution parameters $\mu$ and $\sigma$.

Our mean estimates of the serial interval range from 2.64 days (B47) to 6.74 days (B43) in the primary clusters. The overall mean across all wave 1 clusters is 4.65 days, and across all wave 2 clusters is 5.17 days, though there is considerable variation between and within clusters, causing considerable uncertainty when combining across them. Figure 5 shows the cluster-specific means and 95% confidence

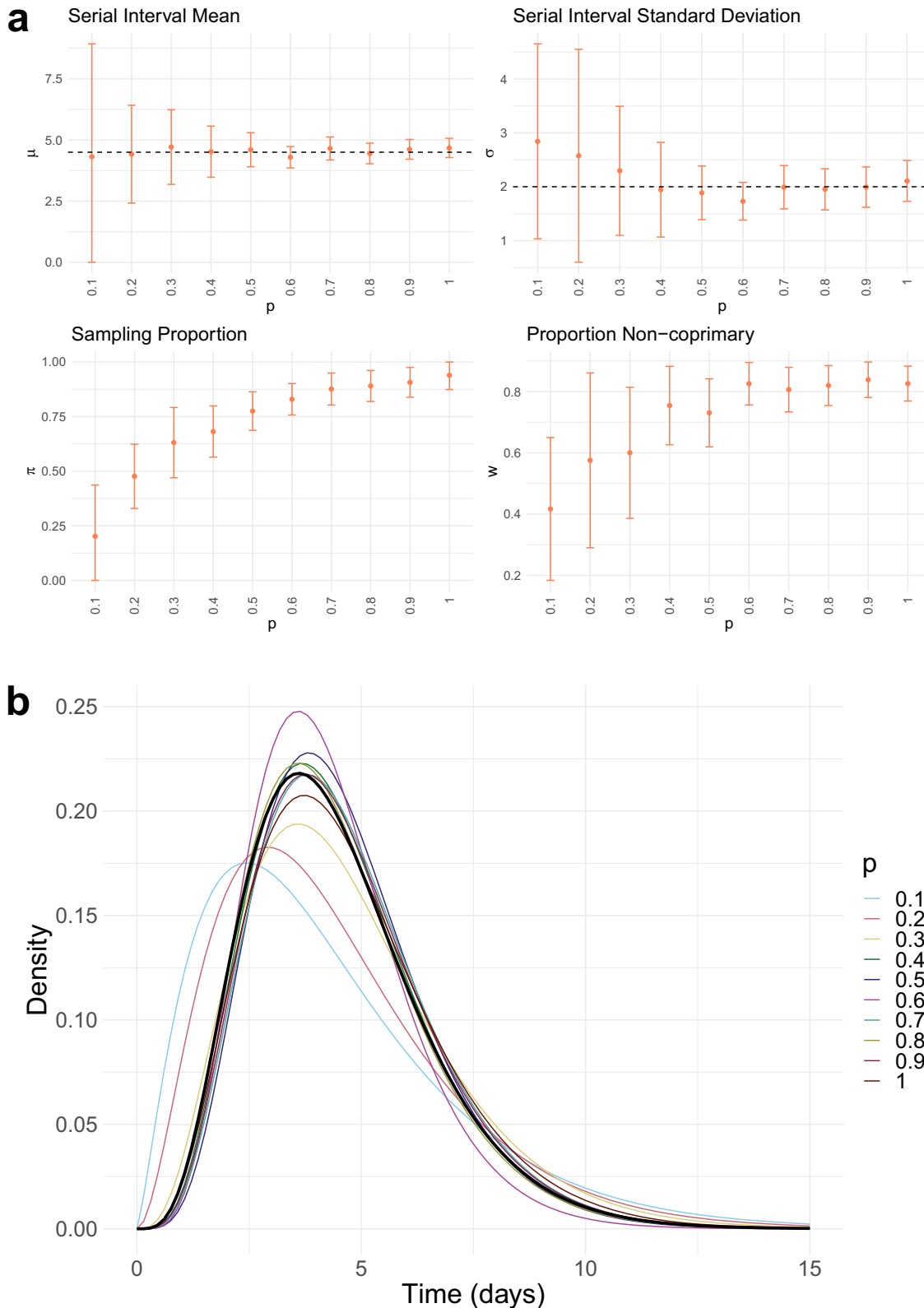

**Fig. 2 | Simulation study results. a** Mean estimates (points) and 95% confidence intervals (bars) of the model parameters, under the varying proportions of sampled cases *p*. Black dashed line shows true parameter values for $\mu$, $\sigma$. **b** Resulting $\Gamma(\mu, \sigma)$ serial interval distributions, calculated from the mean estimates of $\mu$ and $\sigma$. The thicker black line shows the true serial interval distribution. 1000 transmission trees were sampled from an outbreak with a final size of $n = 807$.

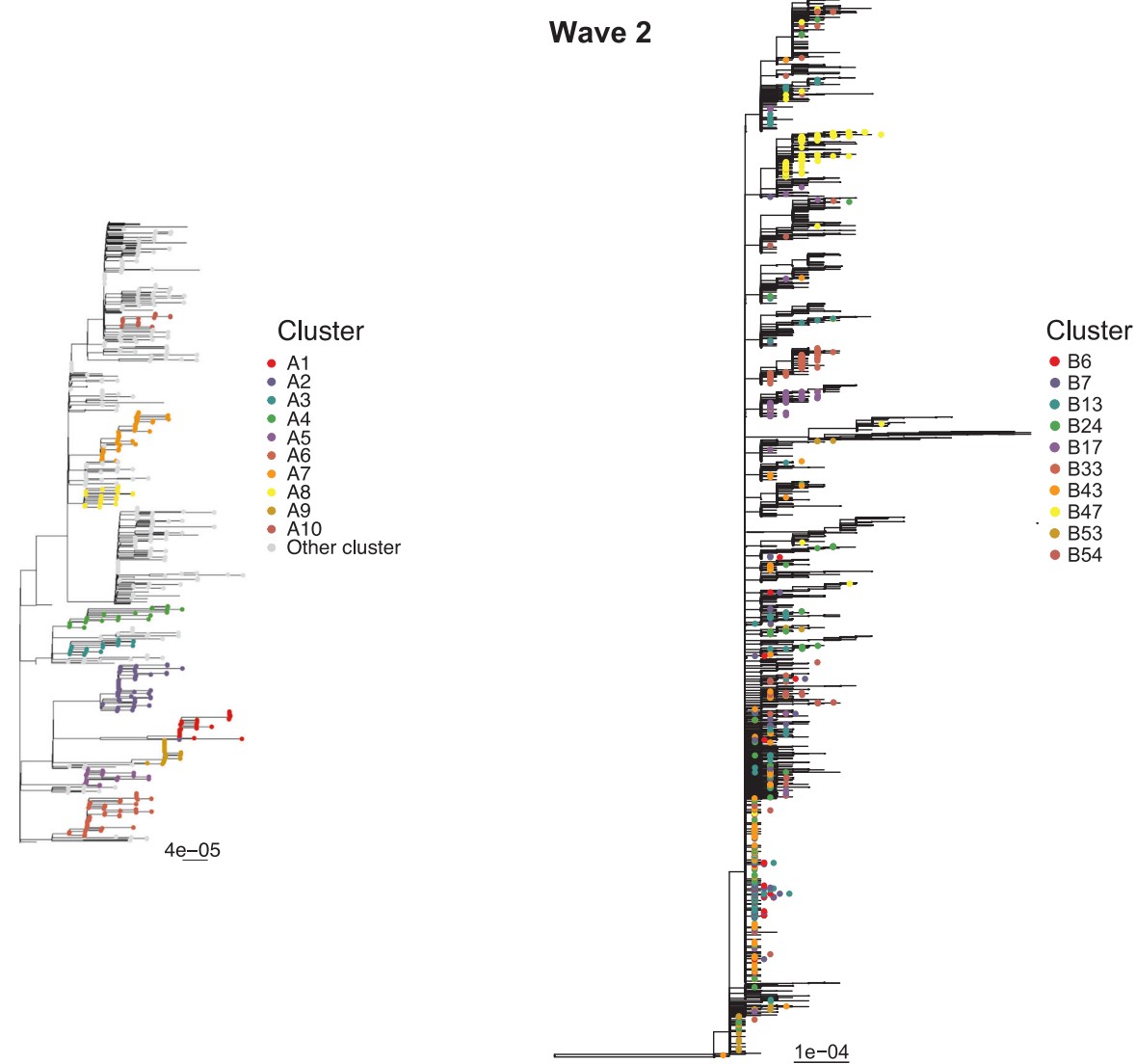

**Fig. 3 | Phylogeny of wave 1 (left) and wave 2 (right) Victorian SARS-CoV-2 data.** Trees are built from the entire set of sequences in each wave, with primary clusters selected for serial interval analysis coloured and labelled. Clusters are defined genomically in wave 1 and epidemiologically in wave 2.

intervals of the model parameters: serial interval mean $\mu$ and standard deviation $\sigma$, the sampling probability within identified transmission chains $\pi$, and the proportion of non-coprimary transmission $w$. Although most of the cluster confidence intervals overlap, those in the aforementioned clusters B47 and B43 are completely disjoint. However, a statistical test for if the 20 primary serial interval means are feasible from the same population serial interval distribution did not reveal any significant difference in our cluster-specific estimates (see Supplemental Materials Section S3). Figure 6 shows the mean gamma-distributed serial interval estimated in each cluster, allowing for a comparison of the mean and standard deviation together.

Full results are included in Table S3 in the Supplemental Materials, presented alongside several published estimates of the COVID-19 serial interval in Table S2 for comparison. Our estimates are not substantially changed by varying the number of transmission networks sampled (Fig. S2), or by preferentially sampling infectors by genomic distance alone rather than genomic and time distance (Table S4). They are in general agreement with others from wild-type (non Variant of Concern [VOC] SARS-CoV-2, though with wider uncertainty (Fig. S14). However, the majority of these published estimates did not take uncertainty in who infected whom, or coprimary/indirect transmission into account.

Estimates of the parameters concerning sampling, $\pi$ and $w$, are controlled relatively strongly by their prior distributions (prior mean 0.52, overall posterior mean 0.59 for $\pi$ and 0.55 for $w$, Fig. 5). However, we do observe some clusters pushing against this, most notably cluster B53, in which the data suggest a lower sampling rate ($\pi = 0.47$). A sensitivity analysis to these assumed prior distributions, included in the Supplemental Materials Section S2, reinforces that $\pi$ and $w$ are influenced by changes to their prior, but our serial interval estimates are robust to moderate changes in these sampling rate priors (on the order of a 20% chance to the prior mean or 50% to the prior standard deviation). We further quantify the influence of the priors on $\pi$ and $w$ by calculating the Kullback-Leibler (KL) divergence between the posterior and prior (the likelihood information) and between the posterior and likelihood (the prior information)[23] for all wave 1 clusters, shown in Figure S10. We again find that the priors are influential on our estimates of $\pi$ and $w$ (though this approach is not able to assess the onward impact to estimating the serial interval), with around 75% of the $\pi$ and $w$ posterior influence coming from the prior and 25% from the likelihood, under our KL statistic. The likelihood is more influential in clusters with a larger number of identified cases (A2, A7), suggesting that with higher genomic surveillance, more data and therefore larger

**Table 1 | Epidemiological details of the primary Victorian SARS-CoV-2 clusters**

| Wave 1 | Cases | Onset date range |
|---|---|---|
| Cluster A1 | 29 | 2020/02/18 - 2020/04/01 |
| Cluster A2 | 61 | 2020/02/28 - 2020/04/02 |
| Cluster A3 | 21 | 2020/02/28 - 2020/04/05 |
| Cluster A4 | 28 | 2020/02/29 - 2020/03/29 |
| Cluster A5 | 19 | 2020/03/08 - 2020/03/31 |
| Cluster A6 | 24 | 2020/03/10 - 2020/04/09 |
| Cluster A7 | 59 | 2020/03/10 - 2020/04/06 |
| Cluster A8 | 29 | 2020/03/10 - 2020/03/28 |
| Cluster A9 | 30 | 2020/03/10 - 2020/04/06 |
| Cluster A10 | 18 | 2020/03/15 - 2020/04/05 |
| Total | 318 | 2020/02/18 - 2020/04/09 |
| **Wave 2** | **Cases** | **Onset date range** |
| Cluster B6 | 20 | 2020/06/14 - 2020/07/26 |
| Cluster B7 | 59 | 2020/06/14 - 2020/07/28 |
| Cluster B13 | 115 | 2020/06/18 - 2020/08/27 |
| Cluster B17 | 142 | 2020/06/27 - 2020/08/19 |
| Cluster B24 | 59 | 2020/07/04 - 2020/08/20 |
| Cluster B33 | 117 | 2020/07/07 - 2020/08/19 |
| Cluster B43 | 111 | 2020/07/12 - 2020/08/02 |
| Cluster B47 | 157 | 2020/07/13 - 2020/09/26 |
| Cluster B53 | 19 | 2020/07/15 - 2020/07/28 |
| Cluster B54 | 44 | 2020/07/15 - 2020/08/29 |
| Total | 818* | 2020/06/14 - 2020/09/26 |

*Note: wave 2 cases may appear in multiple clusters.

clusters, our estimates would be less reliant upon a predetermined prior sampling distribution.

### Estimation of effective reproduction number $R_t$

To contextualise the effects of using cluster-specific serial intervals in downstream analyses, we compare estimates of the effective reproduction number $R_t$ (also known as the time-varying reproduction number) using our cluster-specific serial intervals and literature-based serial intervals. $R_t$ can be interpreted as the expected number of secondary cases caused by an index case at time $t$, and its calculation relies on the assumption of an underlying serial interval distribution. We use the R package EpiEstim[24] to estimate $R_t$ on a weekly sliding window in each cluster, using both the cluster-specific serial interval distribution estimated in this work and a $\Gamma(\mu = 6.3, \sigma = 4.2)$ distribution as estimated for SARS-CoV-2 by ref. [25].

Results of the $R_t$ comparison are shown in Fig. 7; any difference in $R_t$ in these figures arises solely from the difference in the underlying serial interval distribution. Although some clusters remain largely unaffected, for some clusters, the estimate of $R_t$ differs by up to a factor of 2 or 3. This occurs primarily in early cluster estimates of $R_t$, highlighting how even small amounts of uncertainty in the underlying transmission model should be treated carefully when obtaining initial estimates. But even towards the middle or end of cluster outbreaks, the choice of the serial interval can be the difference between an estimate of $R_t < 1$ and $R_t > 1$ (e.g. A10), which has clear implications for epidemic control.

### Serial intervals by the exposure site category

We estimate the serial interval in the full set of 94 wave 2 clusters, categorised by exposure site type and shown in Fig. 8. Estimates of the mean serial interval range from 1.97 to 9.54 days, though many of the largest estimates have considerable uncertainty. We have limited

observations for several exposure site types, but the results suggest some patterns by exposure type beginning to emerge: with meat packing/meat processing plants and schools among the shorter serial intervals and healthcare facilities and housing among the longer. Aged care facilities have the widest range of mean serial intervals, with one cluster in particular (B82) having significantly shorter mean serial interval than the rest (Supplemental Materials Section S3). In the supplementary analysis, our statistical test reveals that packing and meat processing plants have statistically significant shorter serial intervals than the other categories (Supplemental section S3 and Fig. S8). As in the primary clusters, sampling proportion $\pi$ and proportion non-coprimary $w$ do not greatly diverge from their prior distributions. There are several exceptions, such as clusters B53, B71 and B82, all with $\pi \le 0.5$, suggesting a lower rate of case acquisition.

### Comparison to contact-defined clusters

The method introduced in this work is not limited to analyses using genomic data. We compare our genomic approach against an analysis that applies the same serial interval estimation procedure to contact-sampled transmission networks from contact-defined clusters. Here, we define case clusters as the connected components of a network created from contact tracing data. Rather than sampling transmission networks using viral sequences, we preferentially sample infector-infectee pairs with known direct contact as indicated by contact tracing. Full details and results are included in the Supplemental Materials, Section S5. We obtain 13 contact-defined wave 1 clusters, of a comparative size to the genomic clusters (Fig. S11).

We find that contact-cluster-specific estimates of the mean serial interval range from 3.02 days to 7.71 days, with an 'all clusters' estimate of 5.04 days (Fig. S12). Although direct comparison is difficult due to the fact that the genomic and contact-based clusters do not entirely overlap, the estimated serial intervals are similar whether we use genomic sequences or contact data to cluster and build the transmission networks. This is especially true in those pairs of clusters which are most similar under the two clustering methods, sharing at least 50% of the same cases; presented here in Fig. 9. We see good agreement among estimates for the serial interval mean and standard deviation. The cluster with the most significant disagreement (C13/A2) was associated with several instances of international travel from different continents, leading to local transmission (cluster 70 in ref. [6]). We estimate that the contact-defined clusters have a higher sampling proportion and lower proportion coprimary than their corresponding sequence-defined cluster. This is logical, given that we used larger prior means for $\pi$ and $w$, as not all contact-traced cases were successfully genomically sequenced.

## Discussion

Estimates of the serial interval are key to understanding disease spread, and underlie estimation of other quantities, such as the time-dependent reproduction number. Current methods for serial interval estimation are best suited to small, contained populations with high sampling, and require detailed contact studies, which can be resource intensive. However, there is a growing demand for wide-scale epidemiological analyses across populations, and as such, these are often undertaken using estimates of the serial interval from small, early outbreaks in a different location or time than the population under study. Serial intervals are known to contract during disease outbreaks[26], as well as be impacted by whether symptom onset is caused by effects of the virus or host immune response[27], but they may also be changed under different pathogen strains, population mixing or control strategies. In this work, we sought to explore cluster-specific estimates of the serial interval within two waves of the COVID-19 pandemic in Victoria, Australia, through the introduction of a new approach using viral sequences in place of detailed contact data.

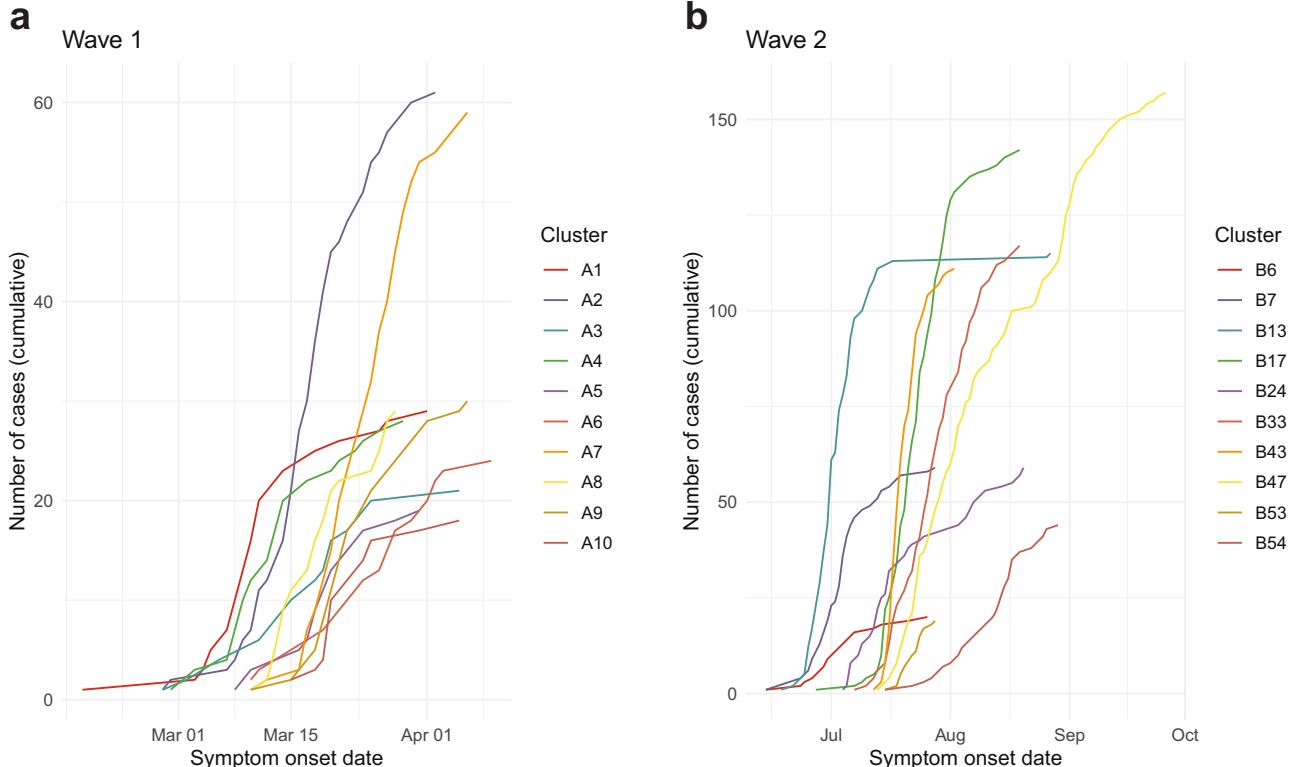

**Fig. 4 | Symptom onset curves of the primary Victorian SARS-CoV-2 clusters. a** Wave 1 and **b** wave 2 clusters. Cumulative observed cases in each cluster are plotted over time, including only those cases with sufficiently high-quality genomic sequences that are used in onward analysis.

Our estimates of the COVID-19 serial interval are similar to those found from detailed contact studies, despite requiring no knowledge of contact between cases and working with incompletely-sampled data. Our results have wider uncertainty than many published estimates, but the majority of these estimates focused on small populations with known contact pairs and did not take potential under-reporting into account. Although there was variation in our 20 primary cluster-specific estimates, this was not found to be statistically significant overall, under a null hypothesis that all cluster-specific mean serial intervals were identical. We found indication, however, that clusters occurring in sites associated with longer-term contact, such as healthcare and aged care, tended to have longer serial intervals than sites attended for shorter lengths of time, such as meat packing or meat processing plants, though more data would be required to confirm this. Although the parameters concerning sampling rate were relatively strongly controlled by a prior distribution, some clusters suggested higher/lower amounts of sampling: such findings could be used to monitor developing outbreaks, especially if the approach was integrated into routine genomic surveillance in real-time. We found that using cluster-specific serial intervals in the estimation of the time-dependent reproduction number as compared to a literature-based estimate changed $R_t$ by up to a factor of 2–3, particularly early in outbreaks. This highlights how variable estimates of the reproduction number can be, particularly when calculated from small outbreaks in specific settings, and suggests caution should be taken when applying existing parameter estimates to analyses of new outbreaks. Although conceptually and methodologically quite different, this approach shares goals with the work of ref. 28 who estimate $R_t$ from viral sequence data using a birth-death skyline model, via estimating time-varying transmission, recovery and sampling rates in BEAST2[29]. Unlike our approach, their focus is on the reconstruction of the phylogeny without consideration of symptom onset times or the serial interval. Nonetheless, an interesting extension could be to compare estimates

of $R_t$ from both methods. Overall, our findings highlight a need for repeated estimation of parameters concerning infection and transmission, both to obtain a consensus and to track whether values are changing in time.

There are several limitations of our methodology. Due to the assumption of gamma-distributed serial intervals (required for the construction of the mixture model), we assume serial intervals are strictly positive. Although this does not preclude presymptomatic transmission, as has been widely noted for COVID-19[11] and can still result in positive serial intervals, there is evidence of negative serial intervals for COVID-19[14] and for other diseases. An extension of our model could allow for this. In our supplementary simulation study, we further found this assumption limits the method's ability to infer serial intervals with mean 1 day. However, this is infeasibly short for many diseases, including COVID-19, see, e.g. the published estimates in Table S2. We perform transmission tree sampling from a cloud of plausible infector-infectee pairs, allowing for indirect and coprimary transmission, in order to take into account uncertainty in who infected whom from the genomic data. Although we do this in a probabilistic way, that aims to approximate the judgement applied in public health (cases closer in pathogen sequence and in time are more likely to infect one another), the tree sampling could be improved by incorporating additional epidemiological data, e.g. known pairs from contact tracing or relative Ct values, if this were available. It may be possible to do this by incorporating existing methodologies for the sampling of transmission trees, such as the outbreaker or TransPhylo platforms[16,17], but these are not well positioned to estimate serial intervals as the underlying transmission models do not consider the time of symptom onset. Lastly, the coprimary transmission model could be extended to include further unsampled intermediate cases, matching the non-coprimary model. This was not a priority in this work, as the high level of sampling makes such scenarios increasingly unlikely, but could be impactful in settings with a larger proportion of unsampled cases. If

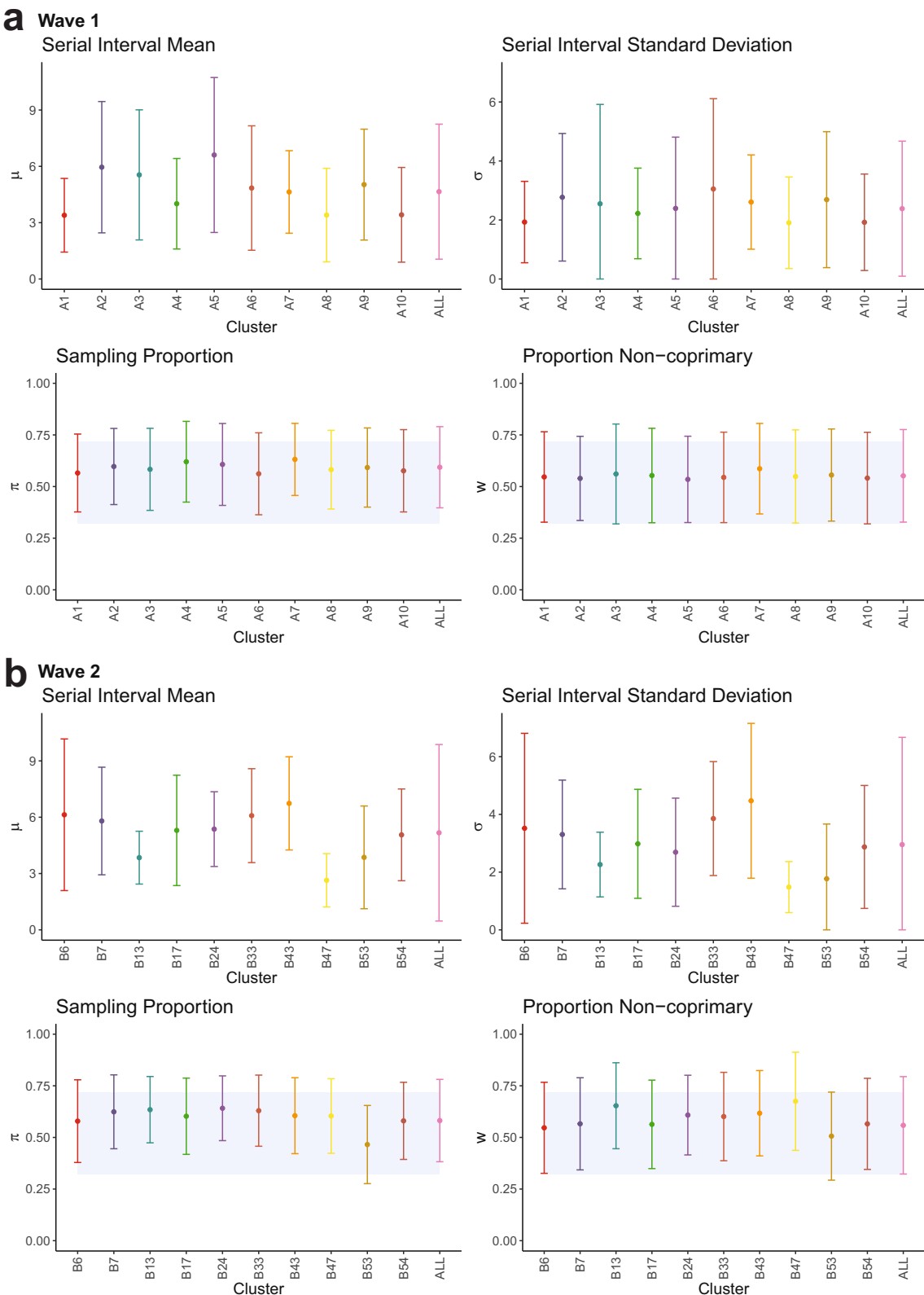

**Fig. 5 | Cluster-specific estimates of model parameters ($\mu$, $\sigma$, $\pi$, $w$).** Results shown for **a** wave 1 and **b** wave 2 clusters, with size as defined in Table 1 and 100 sampled transmission networks. Mean estimates are shown as points and 95% confidence intervals as bars. Blue-shaded areas display the 95% highest density region of the prior distributions on $\pi$ and $w$. Clusters are labelled chronologically by the date of earliest symptom onset. `All' result in wave 2 is calculated from the full set of 94 clusters.

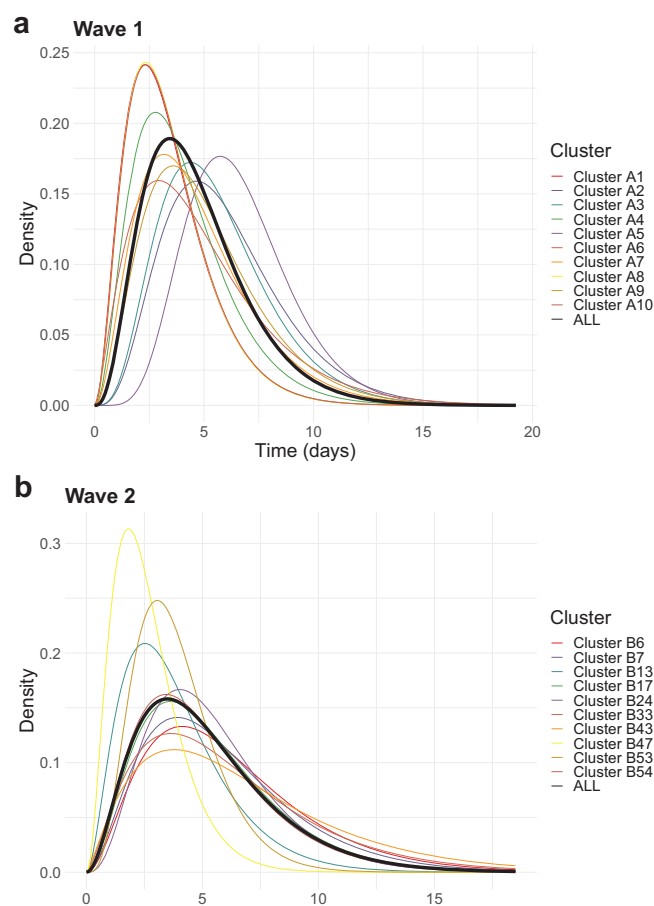

**Fig. 6 | Cluster-specific estimated serial interval distributions.** Results shown for **a** wave 1 and **b** wave 2 clusters. Serial intervals are Gamma distributed, calculated from mean estimates of $\mu$ and $\sigma$. `All' result in wave 2 is calculated from the full set of 94 clusters.

such settings did occur in the data studied here, in reality, the impact would be that our serial intervals are overestimated.

Our approach requires prior knowledge of the population sampling rate. This ties in to an innate identifiability problem with the estimation of serial intervals using any model, in that short serial intervals with low sampling may be indistinguishable from long serial intervals with high sampling. However, the sensitivity analysis of our prior assumptions revealed that our serial interval results are robust to moderate deviations from the sampling rate priors (approximately, mean ± 20%, standard deviation ± 50%). Nonetheless, the sampling-related prior distributions for $\pi$ and $w$ should be chosen carefully, using knowledge of surveillance and sequencing in the population of interest. Due to the potentially complex relationship between the proportion of cases sequenced and the parameters $\pi$ and $w$ (also affected, e.g. by population structure), our simulation study exploring prior choice in section S2 suggested that diffuse priors may be preferable. Another broad challenge in estimating serial intervals is how to incorporate asymptomatic cases, as these individuals naturally will not have a time of symptom onset. In this analysis, we removed all cases with no symptom onset time, and so our serial intervals can be thought of as representing transmission to symptomatic cases, with potentially-asymptomatic unsampled intermediates. Although asymptomatic cases have been identified as transmitters of SARS-CoV-2[30] albeit at a reduced rate, many studies have not differentiated between true asymptomatic cases and cases who were asymptomatic at the time of transmission

but later developed symptoms (presymptomatic, these individuals would be included in our analysis).

A major benefit of the methodology we have presented here is that genomic data can offer a high-resolution view of transmission at a large scale, where collection of contact-tracing data may be expensive or infeasible. During the COVID-19 pandemic, many public health labs undertook routine sample collection and sequencing from a high proportion of identified COVID-19 cases. Whereas, contact tracing teams can be overwhelmed by high caseloads, and contact data collection/sharing is challenging. Our approach could be used beyond serial interval estimation, for example, to compare differences in transmission of COVID-19 VOC, between settings, or between times under particular non-pharmaceutical interventions (NPI). If used in real-time, it could suggest clusters to focus resources upon, for example, those which suggest a lower sampling rate, more rapid transmission, or substantially different or uncertain serial intervals. Our method is not restricted to use on small clusters: one could estimate the population level serial interval, although this would be more computationally demanding. Although reconstruction of the true transmission chain is not our primary aim, rather we consider the space of plausible chains, there is also an opportunity to further explore the genomically-defined sampled transmission networks: for example, to consider patterns of transmission by age, vaccine status or other factors, and whether these change over time. As indicated by the secondary contact-based analysis, the estimation model presented here, which is novel in and of itself, is not limited to situations in which genomic data are used to identify potential pairs. If contact data is available, it can be used to build or inform the collection of feasible transmission networks, and thereby estimate the serial interval distribution. Our estimation model extends the work of ref. 10 by allowing for any number of unsampled intermediate cases and removing the focus on the cluster index case. In this work, we explored how either genomic or contact data alone can teach us about transmission, but in practice, a combination of data sources may result in the best estimates.

More widely, this research enhances the contributions that virus sequences can make to understanding transmission dynamics. To date, phylodynamics has typically operated at the large scale of global phylogeography[31,32] and estimation of the past population dynamics of pathogens at the whole-population scale[33,34]. Genomic epidemiology has, in contrast, had a high level of focus on establishing who infected whom or otherwise analyzing person-to-person transmission[16,17]. Our work establishes an intermediate regime for genomic epidemiology: transmission analysis at the level of small to intermediate groups, in settings where there is insufficient information to identify individual transmission events with high confidence. Our method behaves well in settings with low transmission divergence (low number of mutations separating transmission pairs)[35], as demonstrated in this work. We would expect stronger performance as the amount of sequence variation increases relative to the length of the serial interval, due to increased resolution in the transmission tree sampling, but conversely, our approach is also applicable in settings with longer serial intervals so long as there is sufficient variation in the sequences. Overall, in order to fully harness genomic information, linkage to epidemiological data is helpful – here, times of symptom onset make the link to serial intervals, and exposure sites help to refine clusters. Particularly when this linkage is done at early stages rather than in a *post hoc* manner, analysis can be automated, relies less on human interpretation, and therefore can be incorporated into routine public health monitoring, in real-time if desired.

## Methods
### Simulation study
In the simulation study, with results presented in Fig. 2, we simulate an influenza-like outbreak using the outbreaker package in R[21], that

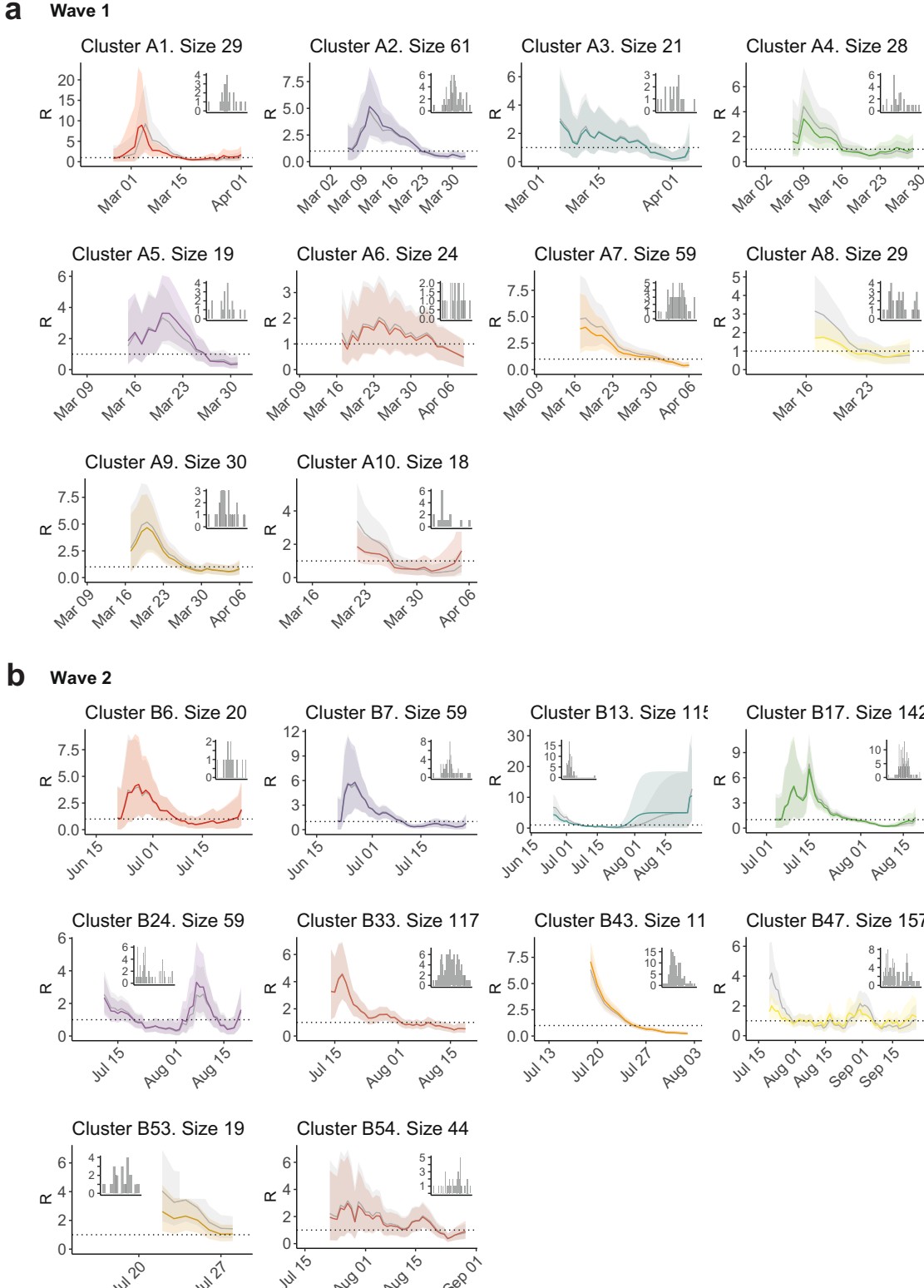

**Fig. 7 | Estimates of the effective reproduction number $R_t$ using cluster-specific serial intervals.** Cluster-specific estimates shown in colour for **a** wave 1 and **b** wave 2. In grey, we compare $R_t$ as estimated using a fixed serial interval distribution, $\Gamma(\mu = 6.3, \sigma = 4.2)$, from ref. 25. Estimates were obtained using the EpiEstim package[24]. Solid lines show posterior mean estimates, and ribbons show 95% credible intervals. Insets show daily incident case counts in each cluster.

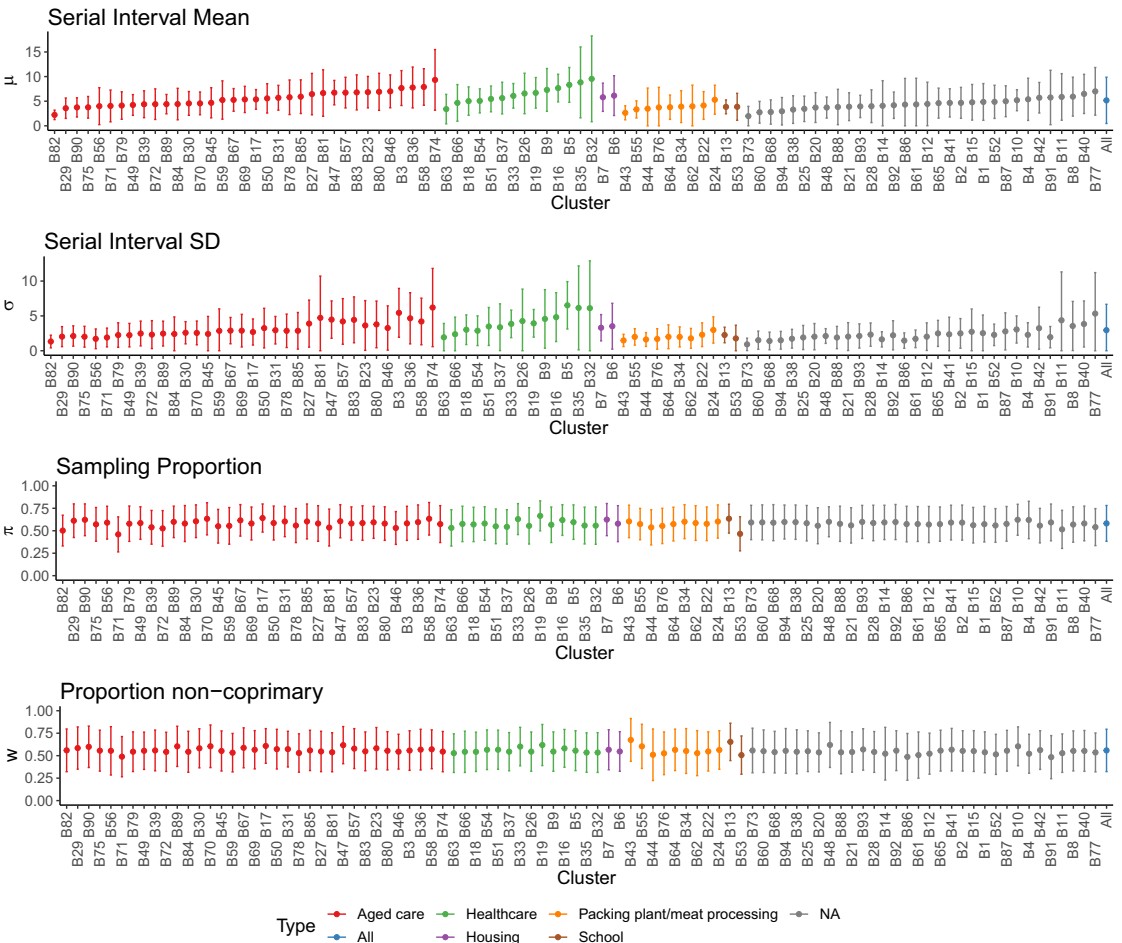

**Fig. 8 | Model parameter estimates for all wave 2 clusters, with size as defined in Table 1 and 100 sampled transmission networks, categorized by exposure site type.** Mean estimates of ($\mu$, $\sigma$, $\pi$, $w$) shown as points, and 95% confidence intervals as bars. Clusters are ordered by Type, and increasing mean serial interval within that.

simulates transmission trees with associated symptom onset times and pathogen genomic sequences. To explore the impact of incomplete case sampling, we run experiments in which we mask an increasing proportion of infected hosts from the serial interval estimation procedure. That is, we run our method with 100%, 90%, ..., 10% of infected cases only, chosen at random. The outbreaker simulator assumes a simple SIR outbreak in which symptom onset and onset of infectiousness occur together, and hence the serial interval is exactly equal to the generation time. We simulate the outbreak with a susceptible population size of 1000, $R_0 = 2$, mutation rate (per site per day) $10^{-4}$ and case importation rate 0.01 per day. The true serial interval distribution used for simulation is $\Gamma(\mu = 4.5, \sigma = 2)$. This results in an outbreak of final size 807.

In the simulation study, we provide our method with the symptom onset time and pathogen genomic sequence from each sampled case, but hide the true transmission tree. For each experiment, in which a proportion $p = 1.0, 0.9, ..., 0.1$ of infected cases are included, we first construct a transmission cloud of feasible transmission pairs, by taking all pairs with genomic distance <0.0001 (see 'Sampling transmission networks' below). This distance is chosen to be approximately as stringent as our Victorian COVID-19 analysis, given the simulated mutation rate. We sample a larger number of transmission networks (1000) than in the Victorian analysis (100), to reflect the larger population size, that may lead to a larger space of plausible networks. From the sampled networks, we obtain MAP estimates of the model parameters ($\mu$, $\sigma$, $\pi$, $w$), using the same approach as for the Victorian COVID-19 outbreaks, described in full below. We use beta distributed prior distributions on $\pi$ and $w$ that represent moderate knowledge of the sampling rate, as this is the situation in which our method is intended to be most suitable. That is, the prior mean is set between 90–110% of included proportion $p$ (varied across experiments), with the prior standard deviation set to 0.1. We note that $p$ is not exactly equivalent to either $\pi$ or $w$, the relationship will depend on the structure of the sampled network. We do not use priors for $\mu$ or $\sigma$.

**Victorian SARS-CoV-2 whole-genome sequencing and clustering**
The remainder of the Methods describes the analysis of the Victorian SARS-CoV-2 clusters. All samples were routinely sequenced as part of public health operations. We provide a brief description of the sequencing and genomic clustering procedure below, for full details, see ref. 6. Whole RNA was extracted from samples obtained from nasopharyngeal swabs, and positive samples for SARS-CoV-2 were identified by RT-qPCR. Positive samples underwent tiled amplicon sequencing using ARTIC primers (v1 and v3)[36,37]. Amplicons were prepared for sequencing on an Illumina sequencer using the NexteraXT library prep protocol following the manufacturer's instructions. Sequencing reads were mapped to the Wuhan-Hu-1 reference sequence (Genbank MN908947.3 [https://www.ncbi.nlm.nih.gov/nuccore/MN908947]) and consensus sequences generated using the iVar pipeline[38]. Consensus sequences were kept for downstream analyses if they met the following criteria: ≥95% genome recovered, ≤25

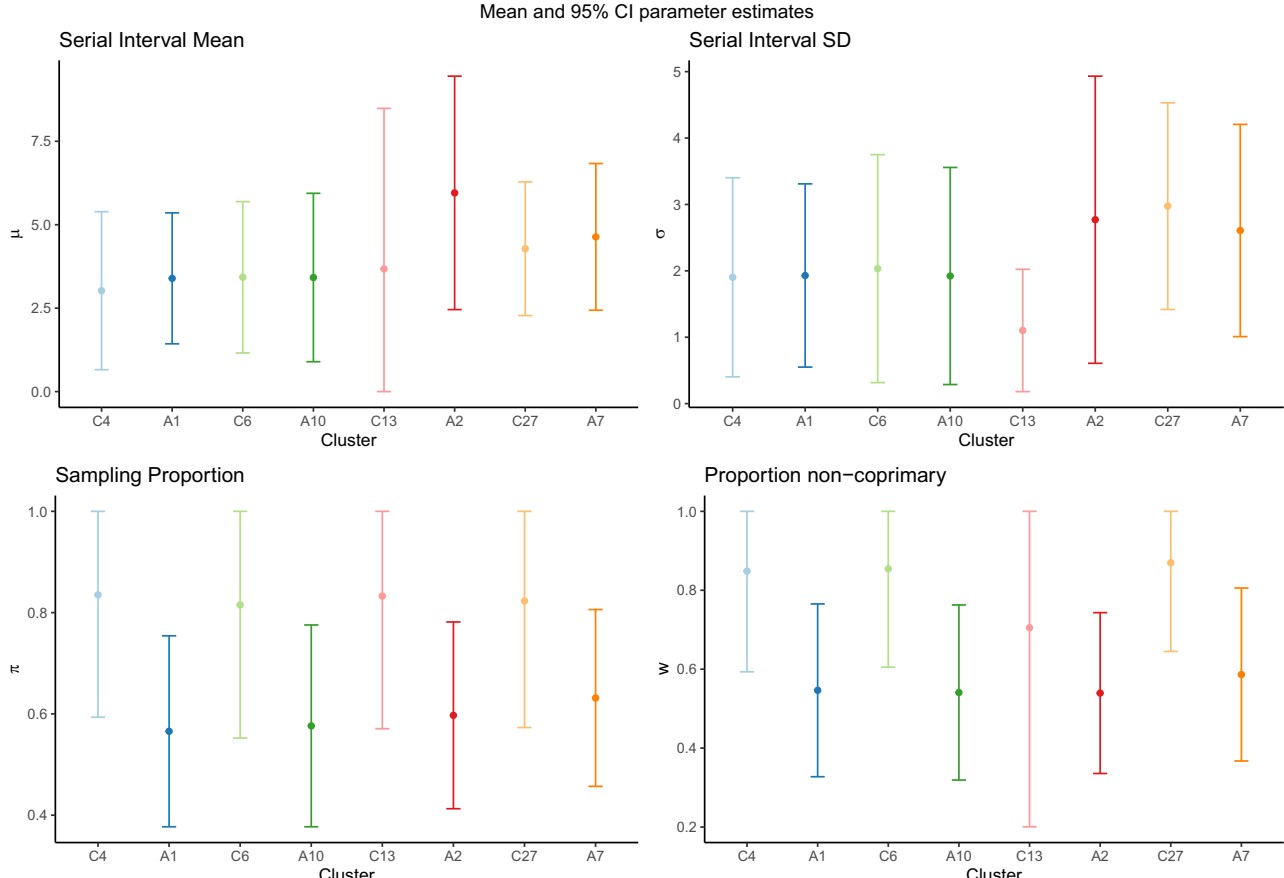

**Fig. 9 | Comparison between model parameter estimates from contact-defined (C#) and sequence-defined (A#) wave 1 clusters, with size as defined in Fig. S11 and Table 1, respectively, and 100 sampled transmission networks.** Mean estimates of $(\mu, \sigma, \pi, w)$ shown as points, and 95% confidence intervals as bars. Each pair of clusters share at least 50% of the same cases: clusters C4 and A10 share ~70% of cases, C6 and A1 share ~67% of cases, C13 and A2 share ~71% of cases, and C27 and A7 share ~51% of cases. The level of sampling $(\pi, w)$ is not expected to be the same in paired genomic and contact clusters.

SNPs from the reference genome, and ≤300 ambiguous bases. The data comprise genomic sequence, sequence sampling date and symptom onset date for each sampled case, in addition to some epidemiological information for wave 2 as described below.

**Wave 1 clustering.** A total 1242 samples from 1075 patients were sequenced during the wave 1 study period: 6 January–14 April 2020. The wave 1 genomic clustering procedure is as described in ref. 6. Of 903/1242 samples passing initial quality control and de-duplication, 737 were identified as belonging to a genomic cluster, for a total of 76 clusters with at least two cases. The clustering of the samples was based on a maximum-likelihood phylogenetic tree containing a single sequence per patient built using IQtree[39] applying a GTR+Γ4 substitution model. Using the ClusterPicker tool[40], clusters were defined as having at least two samples with the inferred ancestral node having at least 95% bootstrap support and the maximum distance within the cluster of 0.0004 expected substitutions/site. The 76 clusters had strong concordance to epidemiologically-defined clusters (a median of 100% of cases in each epidemiologically-linked group were in a single genomic cluster, but genomic clusters were wider: with a median of 43% of cluster cases in a single epidemiological group)[6]. We focus on all genomic clusters with at least 15 cases for the serial interval analysis. After removing sequences with <95% genome recovered or with missing symptom onset time (three cases), the wave 1 data comprise a total of ten wave 1 clusters containing a total of 312 cases.

**Wave 2 clustering.** Even when pathogen samples are routinely collected, genomic clustering can have computational challenges and lead to large, uncertain clusters, particularly when the transmission is widespread in a community and many cases are sampled. In this context, the concept of what constitutes a cluster is less clearly defined, and tree- or sequence-based clustering can lead to infeasibly large clusters. This is especially true for SARS-CoV-2, for which sampled genetic diversity has been described as remarkably low[41], even more so during periods of lockdown and limited international travel[42], leading to challenges in phylogeny building and sub-classification[43]. Public health experts may also be interested in transmission not simply among genetically linked isolates, but within particular locations or groups of people, for example, schools and hospitals. Driven by these factors, for the wave 2 analysis, we derive clusters epidemiologically rather than genomically: choosing all cases associated with particular exposure sites defined by public health.

A total 15,665 samples from 14,075 patients were sequenced during the wave 2 study period: 1 June–28 October 2020. Epidemiological data, including detailed demographic, risk factor and contact tracing data, were collected for each case through an interview conducted by the Victorian Department of Health, and these data were used to determine locations of possible transmission, known as exposure sites. In our primary analyses, we use only the exposure site information (which cases attended which sites) to define the wave 2 clusters, we do not use any demographic or contact data. We perform a secondary analysis in which we re-estimate the serial intervals without the viral sequences but given full knowledge of the contact data.

Of the 10,642/15,665 samples passing quality control and de-duplication, 7116 were identified as associated with at least one exposure site. Of these samples, we remove 1371 cases with missing

symptom onset time. This provides a total of 623 clusters with at least two cases. Recall that cases may be associated with more than one exposure site, and so may be included in more than one cluster. In our primary analysis, we focus on ten exposure site clusters—we select the largest two clusters associated with each of the aged care facilities, healthcare facilities, housing, schools and meat packing/meat processing plants. We perform additional analysis on all wave 2 clusters with at least 15 cases. We remove a further 2 clusters which contain more than 15 cases but have a limited signal of transmission (five or fewer plausible infector-infectee pairs). This results in a total of 94 wave 2 clusters comprised of a total of 3875 cases.

**Phylogenetic reconstruction.** Phylogenies are produced from the entire set of sequences in each wave, using a custom workflow for building fast SARS-CoV-2 trees available at github.com/MDU-PHL/kovid-trees-nf. Briefly, sequences are cleaned to remove sites with >5% missing calls and de-duplicated with GOALIGN[44]. An approximate maximum-likelihood tree is built using FastTree[45] and the branch lengths are optimised with RAxML-NG[46]. Finally, GOTREE[44] is used to repopulate the tree with duplicate sequences.

### Sampling transmission networks

In order to estimate cluster-specific serial intervals from viral sequences and symptom onset times in a cluster (i.e. in the absence of contact tracing data), we require an approach for obtaining putative transmission pairs. Given that there will usually be considerable uncertainty in who infected whom, we sample a set of plausible transmission networks, accounting for indirect transmission and perform parameter estimation across the entire set.

We build a pairwise genetic distance matrix between all aligned sequences in a cluster using the *ape* package in *R* and the TN93 model of evolution (this was found to generate very similar distances to the GTR+Γ4 model used to generate the wave 1 clusters, see Supplemental Materials Fig. S1). We then create a 'transmission cloud', that is, a set of all plausible transmission pairs in the cluster. We define a plausible (possibly indirect) transmission pair as two cases in the same cluster who:

1. have observed interval between symptom onsets $0 < T \leq 35$ days
2. have pairwise genomic distance $G < 1.1/29903$ (where 29,903 is the alignment length).

We apply a much more stringent genomic distance criterion (corresponding to ~2 generations of infection) than the onset distance criterion (corresponding to ~7 generations), to allow the genomic data to be relatively more informative than the time-based data, and to minimise input bias towards short serial intervals. Note that our approach may result in a case having multiple plausible infectors, or no plausible infectors: in the second scenario, the case would be considered as an importation to the cluster from an unsampled case.

From the transmission cloud, we sample a plausible transmission network by sampling an infector for each infectee from among their list of plausible infectors. For infectee $j$, a plausible infector $i$ is selected with higher probability if the genomic distance and/or the difference in symptom onset time between $i$ and $j$ is lower. The sample weighting $s(i, j)$ is given by

$$s(i,j) = \frac{|G_{i,j} - G_{\max}|}{G_{\max}} + \frac{|T_{i,j} - T_{\max}|}{T_{\max}}, \tag{1}$$

where $G_{i,j}$ is the genomic distance between $i$ and $j$, and $T_{i,j}$ is the difference in symptom onset time between $i$ and $j$. $G_{\max}$ and $T_{\max}$ are the maximum genomic and time distance, respectively, among all plausible pairs in the cluster; their inclusion causes distances $G_{i,j}$ and $T_{i,j}$ close to zero to result in higher sampling probability $s_{i,j}$. The weighting is normalised to sum to 1 for each infectee $j$. We repeat this process

$N = 100$ times to obtain a set of 100 plausible transmission networks per cluster.

As we do not assume that all cases have a sampled ancestor within their cluster, a sampled transmission network is not required to be comprised of a single transmission tree; there can be several distinct sub-trees, each spawned by an unsampled case. The result is that we only perform serial interval estimation on those inferred transmission pairs with sufficient confidence.

### Serial interval estimation

After sampling a set of plausible transmission networks per cluster, we proceed to estimate the serial interval in each network, and finally combine these for a single cluster estimate. Our approach seeks to estimate the parameters of the true underlying serial interval distribution, that is, the serial interval arising from direct transmission between a pair of individuals. However, we account for the fact that observed serial intervals in data with incomplete sampling may comprise a mixture of direct transmission ($i \rightarrow j$), indirect transmission ($i \rightarrow x \rightarrow j$, for any number of unsampled $x$) and coprimary transmission ($x \rightarrow i, x \rightarrow j$, for unsampled $x$).

In this description, we assume that a plausible transmission network has already been sampled from the data as above, and so transmission between a sampled infector-infectee pair ($i, j$) is considered certain (even if that transmission may be direct, indirect or coprimary). We apply the methodology described in this section independently for each sampled network, in order to incorporate uncertainty in the network.

Let $T'$ be the true serial interval distribution of the disease under consideration, so that $T'_{ij}$ is the serial interval arising from direct transmission between any case $i$ and case $j$. We assume that $T'_{ij} \sim \Gamma(\mu, \sigma)$, for mean $\mu$ and standard deviation $\sigma$. Note that we assume, therefore, that the serial interval is strictly non-negative. This is the quantity we are interested to make inference about, by estimation of $\mu$ and $\sigma$.

Now, let $T_{ij}$ denote the observed time interval between the symptom onsets of a particular case $i$ and case $j$. If there is direct transmission from $i$ to $j$ then $T_{ij} = T'_{ij}$. However, if we allow for under-reporting in the data, i.e. there may have been unsampled intermediate cases between $i$ and $j$, then $T_{ij}$ is the convolution of multiple serial intervals. It is also possible that both $i$ and $j$ were infected by the same unsampled host, which we call coprimary transmission. Although we allow for a theoretically infinite number of unsampled intermediate cases between cases $i$ and $j$, in the coprimary case, we assume that a single unsampled host must have infected both $i$ and $j$.

More formally, we assume that the serial interval distribution is a mixture of two transmission paths: coprimary and non-coprimary (direct or with unsampled intermediates).

**Non-coprimary transmission.** Sampled infector $i$ and infectee $j$ are separated by $m$ unsampled intermediate hosts, $0 \leq m < \infty$ ($m = 0$ corresponds to direct transmission). Then, the observed interval $T_{ij}$ is the sum of $m + 1$ i.i.d. gamma-distributed intervals, $T_{ij} \sim \Gamma((m+1)\mu, \sqrt{(m+1)}\sigma)$. The value of $m$ is unknown but can be considered to follow a Geometric distribution with success probability $\pi$, where success corresponds to sampling an infected individual and $m$ is the number of failures. The parameter $\pi$ then represents a pseudo-sampling probability, for hosts descending from a sampled infector only. Taking this into account, the marginal distribution of $T_{ij}$ will be a Compound Geometric Gamma distribution, CGG($\mu, \sigma, \pi$). We assume that a proportion $w$ of observed serials are of this type.

**Coprimary transmission.** In the coprimary case, sampled infector $i$ and infectee $j$ were in actuality both infected by the same unsampled case $x$. Since the serial intervals $T'_{xi}$ and $T'_{xj}$ are i.i.d. $\Gamma(\mu, \sigma)$, the observed interval $T_{ij}$ is the strictly non-negative difference of two i.i.d. gamma distributions, $T_{ij} \sim |T'_{xj} - T'_{xi}|$. We refer to this as a Folded

Gamma Difference distribution FGD($\mu, \sigma$). This is a folded version of the Gamma Difference distribution introduced in refs. 47,48, with the simplification that both gamma distributions have the same parameters. The remaining proportion of $1 - w$ of observed serials are of coprimary type.

### Likelihood

We wish to write down the likelihood of the model described above, given a dataset $\mathcal{D}_{c,\tau} = \{T_{i_1 j_1}, T_{i_2 j_2}, \ldots, T_{i_n j_n}\}_{c,\tau}$ which contains the observed time intervals for all $n$ sampled infector-infectee pairs in network $\tau$ for cluster $c$. Note that any given host can be an infector of multiple cases and an infectee as well, so some $i$ and $j$ may refer to the same individuals.

The log-likelihood of the model parameters given dataset $\mathcal{D}_{c,\tau}$ is given by:

$$l_{c,\tau}(\mu,\sigma,\pi,w|\mathcal{D}_{c,\tau}) = \sum_{k=1}^{n} \log\Big(w \times f_{CGG}(T_{i_k j_k}|\mu,\sigma,\pi) \\ + (1-w) \times f_{FGD}(T_{i_k j_k}|\mu,\sigma)\Big), \quad (2)$$

where $f_{CGG}$ and $f_{FGD}$ are PDFs of the Compound Geometric Gamma and Folded Gamma Difference distributions, respectively.

### Maximum a posteriori (MAP) estimation

Rather than maximising the likelihood expression directly to obtain estimates of the model parameters, we take a Bayesian approach to incorporate prior information on parameters $\pi$ and $w$, and hence find maximum a posteriori (MAP) estimates. Given that there may be considerable uncertainty on the true transmission network as obtained from viral sequences and symptom onset times alone, but we often have good prior knowledge of the level of sampling in the population, the use of priors can help to avoid issues of identifiability in the model. We can imagine that, without the restriction of such priors, extremely short serial intervals with very low levels of sampling would provide a good fit to any data, even if we think this is impossible in practice.

We assume a beta distributed prior for both $\pi$ and $w$, desirable as it is restricted to the range [0, 1]. The log prior distributions for $\pi$ and $w$ are given by:

$$q(\pi) \sim \log(\operatorname{Beta}(\alpha_\pi, \beta_\pi)) \quad (3)$$

$$q(w) \sim \log(\operatorname{Beta}(\alpha_w, \beta_w)). \quad (4)$$

Given these prior distributions and the log-likelihood expression introduced in Equation (2), the log posterior distribution is given by:

$$p_{c,\tau}(\mu,\sigma,\pi,w|\mathcal{D}_{c,\tau}) = l_{c,\tau}(\mu,\sigma,\pi,w|\mathcal{D}_c) + q(\pi) + q(w). \quad (5)$$

MAP estimates $(\hat{\mu},\hat{\sigma},\hat{\pi},\hat{w})_{c,\tau}$, conditional on a sampled transmission network $\tau$ for cluster $c$, are then found by maximising the log posterior density in Equation (5). In practice, we numerically optimise the log posterior density using the optim function in R.

What remains is to incorporate uncertainty in the transmission network $\tau$. Rather than sampling a single transmission network $\tau$ from the transmission cloud of potential infector-infectee pairs for cluster $c$, we sample a set of networks $\tau_1$, $\tau_2$, ..., $\tau_N$. MAPs $(\hat{\mu},\hat{\sigma},\hat{\pi},\hat{w})_{c,\tau_i}$ are obtained for each network $\tau_i$ independently. When fewer than $N$ MAP estimates are returned from the optimisation, as can occur when networks are randomly sampled which are not concordant with the assumed priors and so cause the numerical optimisation to fail, we sample additional networks until $N$ MAPs are obtained.

The overall MAP for each parameter in cluster $c$ is then calculated as the mean across all sampled transmission networks, for example:

$$\hat{\mu}_c = \frac{1}{N}\sum_{k=1}^{N}\hat{\mu}_{c,\tau_k}, \quad (6)$$

and similarly for $\hat{\sigma}_c, \hat{\pi}_c$ and $\hat{w}_c$.

### Confidence Intervals

We obtain estimates of the standard error ($\hat{se}$) of MAPs $(\hat{\mu},\hat{\sigma},\hat{\pi},\hat{w})_{c,\tau}$, for a sampled network $\tau$, using the inverse negative Hessian evaluated at the MAPs, as obtained from the numerical optimisation procedure. At confidence level $\alpha$, this provides an approximate confidence interval for $\hat{\mu}_{c,\tau_k}$ of

$$\hat{\mu}_{c,\tau_k} \pm z_{\alpha/2}\, \hat{se}(\hat{\mu}_{c,\tau_k}), \quad (7)$$

and similarly for $\hat{\sigma}_{c,\tau_k}, \hat{\pi}_{c,\tau_k}$ and $\hat{w}_{c,\tau_k}$.

In order to obtain confidence intervals for the overall cluster estimates $\hat{\mu}_c, \hat{\sigma}_c, \hat{\pi}_c, \hat{w}_c$ i.e. over the space of all sampled transmission networks, we must take into account variation both within and between estimates. The variance of the estimator $\hat{\mu}_c$ (and equivalently $\hat{\sigma}_c, \hat{\pi}_c$, and $\hat{w}_c$) is derived with the law of total variance:

$$\hat{\operatorname{Var}}(\hat{\mu}_c) = \mathbb{E}_\tau\Big(\hat{se}(\hat{\mu}_{c,\tau_k})^2\Big) + \operatorname{Var}_\tau\Big(\hat{\mu}_{c,\tau_k}\Big). \quad (8)$$

So, the first term incorporates the average uncertainty in each estimate of the MAP (for each sampled transmission network), and the second incorporates the estimate's variability between sampled transmission networks.

This induces a confidence interval for the cluster-level estimates of

$$\hat{\mu}_c \pm z_{\alpha/2}\sqrt{\hat{\operatorname{Var}}(\hat{\mu}_c)}. \quad (9)$$

### Inclusion and ethics

Data were collected in accordance with the Victorian Public Health and Wellbeing Act 2008. Ethical approval was received from the University of Melbourne Human Research Ethics Committee (study number 1954615.3).

### Reporting summary

Further information on research design is available in the Nature Portfolio Reporting Summary linked to this article.

## Data availability

The data, comprised of GISAID accession numbers, originating/submitting laboratories, symptom onset dates and cluster identifiers for all samples used in this study, are available at github.com/jessicastockdale/genomicSIs[20]. Acknowledgments to the submitting laboratories for the GISAID sequences are available in Supplementary Data 1. Wuhan reference genome, Genbank MN908947.3 [https://www.ncbi.nlm.nih.gov/nuccore/MN908947], was used in sequence mapping. Contact-tracing data used in the supplementary analysis is collected by the Victorian Department of Health and Human Services under legislation and is not publicly available to protect participant privacy. Further release of data is subject to approval by the data custodian, and completion of a data use agreement which may have restrictions on publication and dissemination of data. For further information and to request data used in this analysis, please contact mdu-general@unimelb.edu.au.

## Code availability

The code is available at github.com/jessicastockdale/genomicSIs[20].

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

## Acknowledgements
With this work, we would like to remember Anders Gonçalves da Silva, who sadly passed away during the late stage of this project, for his thoughtful contributions and for bringing our team together. We thank the public health, clinical and microbiology staff across Victoria who have been involved in the testing, clinical care and public health responses to COVID-19. We gratefully acknowledge the originating and submitting laboratories of sequences deposited in the GISAID Database that were used in this study. References to these laboratories are available in Supplementary Data 1. Figure 1 was created with BioRender.com. This work was funded by the Victorian Government (A.G.d.S., C.M., N.L.S. and C.R.L.), who also funded the sequencing costs. Funding was also received from the National Health and Medical Research Council (NHMRC) Australia, through the Medical Research Future Fund: Coronavirus Research Response: 2020 Tracking COVID-19 in Australia using Genomics Grant Opportunity MRF9200006 (B.P.H. and A.E.W.) and NHMRC partnership grant: An evidence-based framework for establishing public health microbial genomics in Australia APP1149991 (B.P.H. and A.G.d.S.). The authors received funding from Natural Science and Engineering Research Council (Canada) Discovery Grants RGPIN-2019-06911 (P.T.), RGPIN-2019-06624 (C.C.), Michael Smith Health Research BC COV-2020-1010 (C.C.) and the Federal Government of Canada's Canada 150 Research Chair programme (C.C.). B.P.H. is supported by an NHMRC Investigator Grant GNT1196103.

## Author contributions
All authors contributed to the study design and conceptualisation. J.E.S., K.S., P.T. and C.C. designed the statistical methodology. J.E.S., K.S., B.S., N.M. and A.G.d.S. wrote the software and ran the analysis: J.E.S. led the serial interval data analysis and supplementary experiments, K.S. developed the simulation study, B.S. and A.G.d.S. led the phylogenetic analyses, N.M. developed the contact data study. P.T. and C.C. supervised and verified the analytical methods. A.G.d.S., A.E.W., N.L.S., C.M., B.P.H. and C.R.L. provided the resources, curated the data (genomic and epidemiological), managed the project and provided public health oversight and interpretation. C.R.L. and C.C. jointly supervised the project. J.E.S. wrote the initial draft and all authors reviewed and edited the manuscript.

## Competing interests
The authors declare no competing interests
