## [Peer Review File · Nature Communications]

Genomic epidemiology offers high resolution estimates of serial intervals for COVID-19REVIEWER COMMENTS

Reviewer #1 (Remarks to the Author):

This paper by Stockdale et al. estimates the phylogenetic-cluster-specific serial interval for SARS-CoV-2. This approach makes sense conceptually and is a potentially useful tool and advancement to genomic epidemiology in general. However, I have some questions about the method and results.

1) The whole approach of partitioning between direct, indirect, and co-prime is unnecessarily convoluted given that the authors have enough data (sequences, sample times, and reasonable estimates of the infectious period for each sample) to sample transmission histories directly, including estimation of the probability of an unsampled infector. Is there some justification for why the full transmission histories were not sampled and then integrated out to estimate γ (μ , σ) directly? This would have the advantage of also reducing the parameter space of model as π would be explicitly handled in the transmission history reconstruction.

2) Why was the set of putative infectors based on a TN93 distance matrix when there already was a GTR+ Γ_4 tree already inferred? This does not make sense as the tree is both a reasonable model for the topology of the true underlying transmission history and a more articulated model for the estimation of the genetic distance.

3) What is the justification for picking the specific genetic distance cut off for the set of potential infectors?

4) I don't understand why potential infectors with a shorter distance to the onset of symptoms were assumed to have a higher sampling weight. Doesn't this assumption just bias the method to artificially short serial intervals by assuming that, for example, for two potential donors with the exact same genetic distance to the infectee, the one whose symptoms started closer to the infectee is a better candidate. Why is this justified rather than say some kind of cut off like if $T_{\{I,j\}} > T_{\{cutoff\}}$ then $s_{\{I,j\}}=0$?

5) The logic for using a different criterion for clusters in the second wave is not well justified beyond that it was too hard to compute the desired quantities with the larger clusters. It's unclear to me why the authors did not only consider clusters that were sufficiently near (in terms of cophentic distance in the IQtree) to a given index tip. Having two completely different criteria for clusters means that two different (but related) quantities are being estimated. Why not set up the same inference for each wave? For example, consider only the set of tips as potential infectors that are within some genetic distance for each tip and estimate the parameters of interest in a hierarchical model where the "transmission location" as the units of the model?

6) It doesn't make sense to have a strong prior on π based on the idea that about 57% of known infection are sequenced. This is probably an overestimate given that a lot of cases are never identified—i.e. the probability of someone being missing is the probability of them being not in the data at all not just not sequenced. It also seems like the result should be very sensitive to this parameter and that the data really don't have any opinion (as makes sense given the analysis) about the magnitude of this parameter. Given that the data are uninformative about this parameter and that the prior seems to be not well justified, I'm not sure that this choice makes sense.

Reviewer #2 (Remarks to the Author):

In "Genomic epidemiology offers high resolution estimates of serial intervals for COVID-19", Stockdale et al. present a novel method for using a combination of viral genomes and symptom onset information to estimate cluster-specific serial intervals for SARS-CoV-2 in Victoria, Australia. The manuscript is clearly written and the motivations behind the work are sound and compelling.

We do need more tools that are able to leverage growing viral genomic resources into actionable epidemiological conclusions. However, my overall enthusiasm for the manuscript, as presented, is low because of a lack of validation, as well as methodological questions and concerns regarding uncertainty in estimates.

Major concerns:

1. My primary concern with this manuscript is that the presented approach for estimating serial intervals has not been adequately validated. In fact, validation is almost entirely missing. The only real attempt at validation that I see is brief references to published serial interval estimates for the same virus in different populations and with more traditional approaches. In other words, the estimates shown here are in line with general estimates from other groups using more traditional approaches (e.g., Table S1 and Figure S7). However, the serial intervals from those published studies vary quite a bit and the confidence intervals for the estimates from this manuscript are quite large. So, it is unclear how meaningful the observed overlap really is.

Additionally, one of the stated advantages of the presented approach is its ability to estimate cluster specific differences in serial interval. The authors show that they do see some differences in their estimates for different clusters but make no attempt to demonstrate that these differences are accurate and epidemiologically meaningful. This is critical to demonstrate.

The authors do have one section that they have titled "Validation", but instead of coming at the beginning of the results (to provide the reader with confidence in the approach prior to presenting results), it comes at the end of results, and it seems more like an extension of their approach rather than true validation. If I am understanding correctly, this "validation" section uses the same underlying methodology that is new and presented in this paper, but it doesn't use genomic data as input, but rather relies on more traditional contact tracing. Given that both estimates rely on the new approach presented in this manuscript, this isn't really validation. Also, I was unable to find any direct comparisons of the estimates from these epi-defined clusters versus the genomic clusters.

Potential options for validation that come to mind:

- Cluster-specific comparison of genomic-based results with traditional, previously published approaches for serial interval estimation
- Random shuffling of sequences and symptom onset dates to better understand signal from the data (i.e., how would the results differ with no true signal)
- Comparison of early outbreaks to later outbreaks driven by VOCs that are documented to have distinct serial intervals
- Analysis of datasets for different viruses (e.g., FluA, Ebola) that have distinct serial intervals, to demonstrate the ability of the method to detect these differences

2. There also needs to be additional validation of specific methodological choices. For example, the choice to sample 100 potential transmission trees. What % of possible trees is this? How would your results change if you sampled more possibilities? Also, the switch from phylogenetic clustering to exposure site-based clustering for wave 1 vs. wave 2 is insufficiently explored. How much of an impact does this have on the resulting estimates? Surely there are strategies (like subsampling or clustering genomes prior to generating phylogenies) that could be used to allow for phylogenetic-based clustering in wave 2.

3. The role of the priors on π (pi) and ω (omega) need to be described in a more quantitative manner in the main manuscript. Based on the data presented in SM, it appears that there is little information in the data to infer the distributions of these two parameters and therefore they are heavily impacted by the choice in prior. Fortunately, it looks like the impact on final serial interval estimates may be more robust, but this is only described in vague terms.

Additional concerns:

1. The beginning of the introduction is not very consistent with the focus of the paper. For example, all of the benefits of WGS data for outbreak investigation are focused on bacterial

outbreaks (e.g., AMR tracking), while the manuscript is focused on viral outbreaks.

2. Line 168: What is meant by "wild-type SARS-CoV-2"? Non variant of concern?
3. Are there also priors on the mean and standard deviation of serial interval, which could be displayed on the figures (e.g., Fig. 4)?
4. Figure 7: It is very difficult to discern any patterns with the way the data are currently displayed. I would suggest testing alternative visualization strategies. For example, sorting clusters by "Type" rather than estimate.
5. Line 235: Clarify the meaning of "within one population." How is population defined?
6. Line 265: Please quantify "moderate deviations."
7. Line 288: "not limited to genomic analyses." This aspect of the paper is not developed in enough detail.
8. Line 298: What is meant by "low transmission divergence"?

Reviewer #3 (Remarks to the Author):

This study estimates the serial interval for COVID-19 cases in Victoria, Australia during the first two waves of the pandemic. The dataset is large, has extensive information from contact tracing, and the study addresses a question of broad interest. The authors employ a novel, probabilistic method of reconstructing transmission chains within outbreak clusters and use it to infer the mean and standard deviation of the serial interval for individual case clusters, and from that derive the distribution of serial intervals for the entire dataset.

Conceptually, the study has two components: reconstruction of the transmission network, and estimation of the serial interval, with the latter dependent on the former. Both parts are (as the authors note) difficult, the first because of undersampling and low genetic diversity at the scale of individual transmissions, the second because the estimated serial interval is highly dependent on the (unknown) degree of undersampling.

My overall conclusion is that the paper is well written and the methods used are promising, but that more should be done to validate the accuracy of those methods.

Two validation methods are described in the paper; both could be made more rigorous:

1) In the first method, some clusters are re-analyzed using only contact tracing data. This exercise yields plausible values for the serial intervals, which is reassuring. It would be better and more persuasive, however, if a direct comparison were made between the estimates from the two methods for each cluster (when such comparison is possible). This would permit a quantitative assessment of their agreement. (Based on the supplemental figure, the agreement is likely to be good.)

2) The second validation method involves varying the two priors (on the non-coprimary transmission fraction and on the sampling fraction). When this is done, the posteriors for those parameters, but not for the serial interval, change substantially, indicating that the data provide little information about the two parameters but a lot of information about the serial interval. This is a surprising result, since the degree of undersampling should be directly reflected in the estimated serial interval. I believe the problem here is that both priors are changed at the same time and in the same direction (assuming I understand the description), leading to at least partial cancellation in the estimate of the serial interval. For example, reducing π from .5 to .4 increases the mean of $(m+1)$ from 2.0 to 2.5, leading to a 20% reduction in the estimate of μ ; reducing w at the same

time increases the fraction assigned to coprimary transmission by 20%, leading to an increase in estimated μ . To really test dependence on the priors, they should be varied independently.

Since this study introduces a novel algorithm for reconstructing transmission networks, the accuracy of that algorithm should also be validated directly. Given the extensive contact tracing data available for these samples, a direct comparison between the assigned probabilities of putative infector/infected pairs and the rate of validation by contact tracing could be done. Since there already exist many algorithms for this kind of reconstruction, it would also be appropriate to provide some text placing this novel approach within the framework of existing methods.

Other issues:

Were there any asymptomatic cases in the dataset? How would these be handled, since the serial interval only applies to symptomatic cases?

While any model requires arbitrary choices, some motivation could be provided for some of the choice. For example, the time window (35 days) allowed for sample pairs to be part of a cluster corresponds to ~ 7 viral generations, while the window in genetic distance (1.1 mutations/genome) is ~ 2 generations. Why the discrepancy? Similarly, the model incorporates 1) a single line of descent with any number of intermediate infections, and 2) co-primary transmission (pairs of samples infected from the same case), but does not allow for coprimary transmission with intermediates. Why the omission -- tractability? What kind of effect might the omission have?

Trivia:

l. 96: Should read 'inversely proportional'.

l. 400: Presumably this should say 'result in higher sampling probability'.

eqn. 1: The last term seems to be missing a μ .

Steve Schaffner

Response to reviewers: NCOMMS-22-11439A

Thank you to the three reviewers for their helpful comments, they have provided valuable improvements to our paper. We have carefully considered the reviewers' comments, and added several new analyses in response. These are detailed point-by-point below. Where changes have been made in the manuscript, line references are provided in the response and refer to the resubmitted version.

Response to Reviewer Comments:

Reviewer #1 (Remarks to the Author):

This paper by Stockdale et al. estimates the phylogenetic-cluster-specific serial interval for SARS-CoV-2. This approach makes sense conceptually and is a potentially useful tool and advancement to genomic epidemiology in general. However, I have some questions about the method and results.

Thank you for your thoughtful questions and comments, we have responded to each below.

1) The whole approach of partitioning between direct, indirect, and co-prime is unnecessarily convoluted given that the authors have enough data (sequences, sample times, and reasonable estimates of the infectious period for each sample) to sample transmission histories directly, including estimation of the probability of an unsampled infector. Is there some justification for why the full transmission histories were not sampled and then integrated out to estimate $\gamma(\mu, \sigma)$ directly? This would have the advantage of also reducing the parameter space of model as π would be explicitly handled in the transmission history reconstruction.

The goal of our method is to estimate the serial interval distribution when it is not possible to sample transmission histories directly, i.e. when we have sequences, sample times and symptom onset times but not estimates of the infectious period - because we do not know when people were infected or when they became infectious. In the absence of infection times, we cannot use literature-based estimates of the infectious period without making assumptions about the relationship between infectiousness onset and symptom onset (e.g. that these times are equal, or have a constant difference). Although for this dataset we do have some knowledge of possible infection times from contact tracing information, we do not have this for all cases and we only use this information for validation - as the intent is for this methodology to be applicable in settings where no contact tracing data is available.

We have added text to clarify the above,

“We present a framework that uses virus sequences in place of direct knowledge of infection pairs, for inference of the serial interval distribution in incompletely-sampled case clusters. Our approach does not restrict the number of unsampled intermediate cases, or make

assumptions about the infectious period or latent distribution. It allows for the possibility of presymptomatic transmission but does assume serial intervals are positive. We incorporate...” line 61

“...we assume serial intervals are strictly positive. Although this does not preclude presymptomatic transmission, as has been widely noted for COVID-19 [11] and can still result in positive serial intervals, there is evidence of negative serial intervals for COVID-19 [14]...” line 311

2) Why was the set of putative infectors based on a TN93 distance matrix when there already was a GTR+Gamma4 tree already inferred? This does not makes sense as the tree is both a reasonable model for the topology of the true underlying transmission history and a more articulated model for the estimation of the genetic distance.

This was in part a practical decision. We have a GTR+G4 tree for the wave 1 data from a previous analysis (not wave 2). Here we chose to use the *dist.dna* function in *ape*, where GTR is not available and TN93 is the closest best model, in order to keep all analysis in *R* and therefore provide a simple, adaptable codebase. TN93 is a reasonable model for viruses and is often used to model viral evolution: the different rates for purines and pyrimidines is reasonable here given the observed mutation rates for different types of mutations in SARS-CoV-2 e.g. A->C. In addition, there is very low diversity between our pairs and very high uncertainty in our sampled trees (one key motivation for our estimation & averaging across trees framework). Choice of substitution model is more important at larger distances, when mutation saturation starts to occur and we want to correct for it.

Nonetheless, we did some exploratory analysis of the substitution model choice. We ran model selection in *IQTree2* on the subset of our sequence alignment corresponding to the primary 10 wave 1 clusters, and found TN93 and GTR+G4 had very similar AIC and BIC; they ranked 10th and 18th respectively among the ~130 models compared. We also directly compared the within-cluster pairwise distances obtained from the TN93 and GTR+G4 models for our wave 1 data, and found that the pairwise distances were very similar under both substitution models; the vast majority were within 1 expected substitution, even more so when we restricted the comparison to those small pairwise distances which we consider to be plausible pairs in our analysis.

We have added the latter analysis to the supplementary information, and referenced it in the main text.

Figure S1, and “...and the TN93 model of evolution (this was found to generate very similar distances to the GTR+Gamma4 model used to generate the wave 1 clusters, see Supplemental Materials Figure S1)” line 490

3) What is the justification for picking the specific genetic distance cut off for the set of potential infectors?

Our method aims to use virus genomes, in a sense, as a proxy for contact data, identifying plausible transmission pairs. We needed to be quite “permissive” in the time interval criterion, because time is the target of inference -- being restrictive would bias the inference towards short times by excluding pairs. If genomes are to act as a proxy for contact data, the

genomic distance cutoff needs to play a strong role in the selection of pairs (as identifying contacts would, if the data came from contact tracing). Accordingly, we set the genomic distance cutoff to a value that aims to be inclusive enough to capture variability in the numbers of polymorphisms that occur in a generation of infection, but restrictive enough that the virus genomes play the majority of the role of identifying plausible transmission pairs.

We have added more explanation to the manuscript:

“We apply a much more stringent genomic distance criterion (corresponding to approximately 2 generations of infection) than onset distance criterion (corresponding to approximately 7 generations), to allow the genomic data to be relatively more informative than the time-based data, and to minimize input bias towards short serial intervals.” line 497

4) I don't understand why potential infectors with a shorter distance to the onset of symptoms were assumed to have a higher sampling weight. Doesn't this assumption just bias the method to artificially short serial intervals by assuming that, for example, for two potential donors with the exact same genetic distance to the infectee, the one whose symptoms started closer to the infectee is a better candidate. Why is this justified rather than say some kind of cut off like if $T_{\{i,j\}} > T_{\{cutoff\}}$ then $s_{\{i,j\}}=0$?

These are good questions. In our view, the decision here is in finding a balance between limiting bias and giving the method sufficient information to work with. We determined that the best choice was to set a lenient onset difference cutoff of 35 days (*line 495*) and then to offer higher sampling weight for closer $T_{\{i,j\}}$ below that, whilst sampling a large set of transmission trees. If we were to only set a lenient cutoff, the symptom onset data would not be able to inform our analysis. But the results would likely be very sensitive to the choice of a more stringent cutoff. Our approach is intended to replicate the kind of decision making made in contact studies where, when all other factors are equal, a potential infector will often be chosen as the case closest to the infectee in time.

We have added a sensitivity analysis in which we remove the symptom onset difference aspect of the sample weighting: instead using only genomic distance to get the weights. We still use the strict upper cutoffs to build the transmission cloud (symptom onset difference <35 days). The results of this can be found in *Table S4*, and referenced in the main text *line 205*. We find that our parameter estimates are broadly the same, but with wider uncertainty.

5) The logic for using a different criterion for clusters in the second wave is not well justified beyond that it was too hard to compute the desired quantities with the larger clusters. It's unclear to me why the authors did not only consider clusters that were sufficiently near (in terms of cophentic distance in the IQtree) to a given index tip. Having two completely different criteria for clusters means that two different (but related) quantities are being estimated. Why not set up the same inference for each wave? For example, consider only the set of tips as potential infectors that are within some genetic distance for each tip and estimate the parameters of interest in a hierarchical model where the “transmission location” as the units of the model?

There are indeed several different ways we could have defined the clusters, but unfortunately most that depend on genetic distance do not solve the issue of low diversity in

the wave 2 clusters. The hierarchical model is an interesting alternative, but would require decisions around what to do when we do not know a transmission location or where clusters overlap (cases associated with more than one location).

Although we used a different clustering procedure in wave 1 and wave 2, the wave 1 genomic clustering was found to agree very strongly with an epidemiological approach, limiting the impact of changing methods between waves. We have added this to the results section:

“Sequences are clustered genomically in the first wave, and epidemiologically in the second wave: the epidemiological clustering procedure does not use any detailed demographic or contact data, we group cases associated with exposure sites defined by public health, including schools, healthcare and workplaces. The genomic clustering procedure in wave 1 was found to have strong concordance with an epidemiological clustering approach for this data [6], with a median 100% of cases in each epidemiologically-linked group being in a single genomic cluster.” line 148

We have added further discussion and explanation in Methods on the switch between genomic and exposure-site clustering,

“Even when pathogen samples are routinely collected, genomic clustering can have computational challenges and lead to large, uncertain clusters, particularly when transmission is widespread in a community and many cases are sampled. In this context, the concept of what constitutes a cluster is less clearly defined, and tree- or sequence-based clustering can lead to infeasibly large clusters. This is especially true for SARS-CoV-2, for which sampled genetic diversity has been described as remarkably low [37], even more so during periods of lockdown and limited international travel [38], leading to challenges in phylogeny building and sub-classification [39]” line 447

The genetic diversity in our wave 2 clusters was very low, noted in the text with reference to the phylogeny in Fig. 3 (*line 169*). Given the length of this manuscript and the fact that the clustering procedure is not our main focus or novel contribution, we did not include a detailed comparison of wave 2 genomic clustering approaches. However, the issues we found with very large genomic clusters have motivated a second paper by a subset of the authors (<https://bmccgenomics.biomedcentral.com/articles/10.1186/s12864-022-08936-4>).

6) It doesn't make sense to have a strong prior on π based on the idea that about 57% of known infection are sequenced. This is probably an overestimate given that a lot of cases are never identified—i.e. the probability of someone being missing is the probability of them being not in the data at all not just not sequenced. It also seems like the result should be very sensitive to this parameter and that the data really don't have any opinion (as makes sense given the analysis) about the magnitude of this parameter. Given that the data are uninformative about this parameter and that the prior seems to be not well justified, I'm not sure that this choice makes sense.

The prior on π incorporates both known infections that were not sequenced (43%) and an estimate of the proportion of cases that were never identified (~10%). It is worth noting that π does not represent the overall level of sampling in the population, but rather the level of sampling *within* the identified transmission chains. I.e., a case has to be *infected by* and an

infector of another case in our dataset (or mid-chain of unsampled cases leading to and from sampled cases) to be included in π . Case follow-up was incredibly detailed in Victoria during the study period. During the period 17-30 Aug '20 for example, of the 92% locally-acquired cases, only 15% did not have an identified source of acquisition [<https://pubmed.ncbi.nlm.nih.gov/32907528/>].

We have added discussion of this to the text,

“This is informed by the proportion of identified Victorian COVID-19 cases during the study period that were sampled and sequenced with sufficient quality (57%) combined with a prior belief of a high case finding rate, motivated by detailed case follow-up and high source-acquisition in Victoria at the time [21]. Note that π represents the within-transmission-chain sampling probability, rather than the overall sampling probability in the cluster or population at large: it only concerns cases that were not identified or sequenced but did infect others within the cluster.” line 182

With regards to the second part of this comment, we agree that the results are sensitive to the choice of prior on π and w , though we would not say that the data do not have any opinion. However, we have added more discussion to highlight to readers that the priors should be chosen carefully based on local population and surveillance knowledge (*line 335*). We have also added a new supplementary analysis (*Section S4*) investigating the role of the π and w priors, using the Kullback-Leibler divergence of posterior->prior and posterior->likelihood to assess the relative influence on the posterior of likelihood & the prior (*Figure S7*). As expected based on our previous results, we find that the priors are influential on our estimates of π and w , quantified to contributing around 75% of our ‘relative posterior influence’ (as defined with the KL divergence - with the remaining 25% from the likelihood), but find that the prior is less influential in larger clusters (down to as much as 50%). This method quantifies the impact of the priors on estimating π and w : the influence on those parameters we are most interested in (μ and σ) would then be lower.

We have added this in the main manuscript in Results:

“A sensitivity analysis to these assumed prior distributions, included in the Supplemental Materials Section S2, reinforces that π and w are influenced by changes to their prior, but our serial interval estimates are robust to moderate changes in these sampling rate priors (on the order of a 20% change to the prior mean or 50% to the prior standard deviation). We further quantify the influence of the priors on π and w by calculating the Kullback-Leibler (KL) divergence between the posterior and prior (the likelihood information) and between the posterior and likelihood (the prior information) [22] for all wave 1 clusters, shown in Figure S7. We again find that the priors are influential on our estimates of π and w (though this approach is not able to assess the onward impact to estimating the serial interval), with around 75% of the posterior influence coming from the prior and 25% from the likelihood, under our KL statistic. The likelihood is more influential in clusters with a larger number of identified cases (A2, A7), suggesting that with higher genomic surveillance, more data and therefore larger clusters, our estimates would be less reliant upon a predetermined prior sampling distribution.” line 213

And noted briefly again in the Discussion.

“However, the sensitivity analysis of our prior assumptions revealed that our results are robust to moderate deviations from the sampling rate priors (approximately, mean +-20%, standard deviation +-50%). Nonetheless, the sampling-related prior distributions for π and w should be chosen carefully, using knowledge of surveillance and sequencing in the population of interest” line 333

Reviewer #2 (Remarks to the Author):

In “Genomic epidemiology offers high resolution estimates of serial intervals for COVID-19”, Stockdale et al. present a novel method for using a combination of viral genomes and symptom onset information to estimate cluster-specific serial intervals for SARS-CoV-2 in Victoria, Australia. The manuscript is clearly written and the motivations behind the work are sound and compelling. We do need more tools that are able to leverage growing viral genomic resources into actionable epidemiological conclusions. However, my overall enthusiasm for the manuscript, as presented, is low because of a lack of validation, as well as methodological questions and concerns regarding uncertainty in estimates.

Thank you for your interest and your helpful review. We have added several new sources of validation to the manuscript, and responded to each comment below.

Major concerns:

1. My primary concern with this manuscript is that the presented approach for estimating serial intervals has not been adequately validated. In fact, validation is almost entirely missing. The only real attempt at validation that I see is brief references to published serial interval estimates for the same virus in different populations and with more traditional approaches. In other words, the estimates shown here are in line with general estimates from other groups using more traditional approaches (e.g., Table S1 and Figure S7). However, the serial intervals from those published studies vary quite a bit and the confidence intervals for the estimates from this manuscript are quite large. So, it is unclear how meaningful the observed overlap really is.

Additionally, one of the stated advantages of the presented approach is its ability to estimate cluster specific differences in serial interval. The authors show that they do see some differences in their estimates for different clusters but make no attempt to demonstrate that these differences are accurate and epidemiologically meaningful. This is critical to demonstrate.

The authors do have one section that they have titled “Validation”, but instead of coming at the beginning of the results (to provide the reader with confidence in the approach prior to presenting results), it comes at the end of results, and it seems more like an extension of their approach rather than true validation. If I am understanding correctly, this “validation” section uses the same underlying methodology that is new and presented in this paper, but it doesn’t use genomic data as input, but rather relies on more traditional contact tracing. Given that both estimates rely on the new approach epi-presented in this manuscript, this isn’t really validation. Also, I was unable to find any direct comparisons of the estimates from these epi-defined clusters versus the genomic clusters.

Potential options for validation that come to mind:

- Cluster-specific comparison of genomic-based results with traditional, previously published approaches for serial interval estimation
- Random shuffling of sequences and symptom onset dates to better understand signal from the data (i.e., how would the results differ with no true signal)
- Comparison of early outbreaks to later outbreaks driven by VOCs that are documented to have distinct serial intervals
- Analysis of datasets for different viruses (e.g., FluA, Ebola) that have distinct serial intervals, to demonstrate the ability of the method to detect these differences

Thank you for these suggestions. Informed by these, we have added further validation of our approach in three main areas:

1. Simulation study, with estimation of serial intervals from simulated genomic data in outbreaks with known serial interval distribution
2. Quantitative analysis of the difference in cluster-specific estimates
3. Comparison of genomic vs contact data clusters

Our approach requires less labour-intensive and hard-to-share data than standard methods, which use confirmed contacts (from contact tracing investigations where there is near-complete sampling), but data availability is still a challenge. Our approach requires virus genomes linked to symptom onset times from relatively-densely-sampled outbreaks, together with permission to publish on this data. Symptom onset times are not frequently published publicly alongside virus genomes. We are working on obtaining data sources for future study and hope that the following three analyses provide increased confidence in our methodology and conclusions:

1. We have added a new Validation section at the start of Results ('Validation against simulated data' *line 128*), where we test our method against a simulated outbreak with known serial interval distribution. We perform 10 experiments in which we downsample 0%, 10%, ..., 90% of the cases in the simulated outbreak, to investigate the impact of under-ascertainment on our parameter estimates.

The new *Figure 2* shows the results of the simulation study in the main text, and is described in *lines 129-143*. Full details are given in Methods ('Simulation study' *line 396*). To summarise: similarly to the Victorian analysis, we find that our method is able to estimate the serial interval distribution well when we have fair but imperfect knowledge of the population sampling rate (we intentionally set priors with slightly incorrect mean, and s.d.= 0.1, in the simulation study to model this imperfect knowledge of sampling). There is wider uncertainty around our estimates when we have a lot of missing data, but in this simulation study we find strong performance when we sample at least 50% of cases (with confidence intervals on the mean serial interval on the order of 1 day width)

To the best of our knowledge there are no published approaches for serial interval estimation that account for unsampled cases, except for the work of Vink et al which is not suitable for our population-level clusters because it considers all cases relative to a single household index case. Validation against current approaches that do not account for unsampled cases would not be comparing like-with-like, and we would not expect the serial interval estimates

to be comparable. However, this simulation study gives further insight to the performance of our method.

2. We have added new analysis that more quantitatively explores the differences in our cluster-specific serial interval estimates - *Section S3* of the supplement and quoted in the main text.

We first test if our 20 primary serial interval estimates feasibly have the same mean. We find insufficient evidence to reject the null hypothesis, and therefore cannot conclude that the 20 primary clusters have different mean serial intervals. We repeat this experiment within each of the exposure site categories under wave 2 (as in *Figure 8*), and draw the same conclusion with the exception of aged care facilities, for which we find cluster B82 has significantly different mean. With the exception of B82, this tells us that serial intervals estimated from the *same* exposure site are observations of the same exposure-site-level mean.

Given this result (and excluding B82), we are able to estimate the value of that overall mean serial interval for each exposure site type, shown in *Figure S6*. Lastly, we repeat the test of identical serial intervals once more, on the results of *Figure S6*, to explore now if *different* exposure sites have different means. We reject the hypothesis of identical means, with $p = 3 \times 10^{-4}$. In particular, we find packing and meat processing plants to have significantly shorter serial intervals than other settings, in particular than aged care and healthcare.

In addition to the description of this analysis in the supplement, we have updated the text to reflect these findings:

Results: "Although most of the cluster confidence intervals overlap, those in the aforementioned clusters B47 and B43 are completely disjoint. However, a statistical test for if the 20 primary serial interval means are feasibly from the same population serial interval distribution did not reveal any significant difference in our cluster-specific estimates (see Supplemental Materials Section S3)" line 196

"We have limited observations for several exposure site types, but the results suggest some patterns by exposure type beginning to emerge: with meat packing/meat processing plants and schools among the shorter serial intervals and healthcare facilities and housing among the longer (Supplemental Figure S6). Aged care facilities have the widest range of mean serial intervals, with one cluster in particular (B82) having significantly shorter mean serial interval than the rest (Supplemental Materials Section S3)" line 246

Discussion: "Although there was variation in our 20 primary cluster-specific estimates, this was not found to be statistically significant overall, under a null hypothesis that all cluster-specific mean serial intervals were identical. We found indication however that clusters occurring in sites associated with longer-term contact, such as healthcare and aged care, tended to have longer serial intervals than sites attended for shorter lengths of time, such as meat packing or meat processing plants and schools, though more data would be required to confirm this." line 294

3. We have added further comparison of our genomic- and contact-defined clusters, in the supplement and main text. A full comparison is difficult, because there is incomplete intersection of the genomic and contact clusters, now highlighted in *Figure S8*. However, we

have introduced a new figure to the main text, *Fig. 9*, which compares the serial interval estimates of those 4 pairs of genomic and contact clusters sharing at least 50% of cases.

This is described fully in the supplement, *Section S5*, and discussed in the main text in Results:

“Although direct comparison is difficult due to the fact that the genomic and contact-based clusters do not entirely overlap, the estimated serial intervals are similar whether we use genomic sequences or contact data to cluster and build the transmission networks. This is especially true in those pairs of clusters which are most similar under the 2 clustering methods, sharing at least 50% of the same cases; presented here in Figure 9. We see good agreement among estimates for the serial interval mean and standard deviation. The cluster with most significant disagreement (C13/A2) was associated with several instances of international travel from different continents, leading to local transmission (cluster 70 in [6]). We estimate that the contact-defined clusters have a higher sampling proportion and lower proportion coprimary than their corresponding sequence-defined cluster. This is logical, given that we used larger prior means for π and w , as not all contact-traced cases were successfully genomically sequenced” line 266

In the new *Figure S12* we also compare the infector-infectee probabilities assigned in our transmission tree building procedure against the known contact pairs. We do this for all infectees for whom contact tracing data is available. With this figure, we can visually explore if potential infectors assigned a high probability in our genomic analysis, tend to also correspond to known contact links or not, and this is the case (*Section S5*).

Finally, we have relabelled this section in the main text, previously called ‘Validation’, as instead a ‘Comparison to contact-defined clusters’ (*line 256*).

2. There also needs to be additional validation of specific methodological choices. For example, the choice to sample 100 potential transmission trees. What % of possible trees is this? How would your results change if you sampled more possibilities? Also, the switch from phylogenetic clustering to exposure site-based clustering for wave 1 vs. wave 2 is insufficiently explored. How much of an impact does this have on the resulting estimates? Surely there are strategies (like subsampling or clustering genomes prior to generating phylogenies) that could be used to allow for phylogenetic-based clustering in wave 2.

We have added an additional sensitivity analysis to explore the impact of the choice of number of trees sampled (*Fig S2*). We found that sampling either 50 or 200 transmission trees gives almost exactly the same estimates as our primary analysis with 100 trees, and we have added this to the Results text:

“Our estimates are not substantially changed by varying the number of transmission networks sampled (Figure S2)” line 203

Although we used a different clustering procedure in wave 1 and wave 2, the wave 1 genomic clustering was found to agree very strongly with an epidemiological approach, limiting the impact of changing methods between waves. We have added this to the results section:

“Sequences are clustered genomically in the first wave and epidemiologically in the second wave: the epidemiological clustering procedure does not use any detailed demographic or contact data, we group cases associated with exposure sites defined by public health, including schools, healthcare and workplaces. The genomic clustering procedure in wave 1 was found to have strong concordance with an epidemiological clustering approach for this data [6], with a median 100% of cases in each epidemiologically-linked group being in a single genomic cluster.” line 148

We have added further discussion and explanation in Methods on the switch between genomic and exposure-site clustering,

“Even when pathogen samples are routinely collected, genomic clustering can have computational challenges and lead to large, uncertain clusters, particularly when transmission is widespread in a community and many cases are sampled. In this context, the concept of what constitutes a cluster is less clearly defined, and tree- or sequence-based clustering can lead to infeasibly large clusters. This is especially true for SARS-CoV-2, for which sampled genetic diversity has been described as remarkably low [37], even more so during periods of lockdown and limited international travel [38], leading to challenges in phylogeny building and sub-classification [39]” line 447

The genetic diversity in our wave 2 clusters was very low, noted in the text with reference to the phylogeny in *Figure 3 (line 169)*. Given the length of this manuscript and the fact that the clustering procedure is not our main focus or novel contribution, we did not include a detailed comparison of wave 2 genomic clustering approaches. However, the issues we found with very large genomic clusters partly motivated a paper on clustering by a subset of the authors (<https://bmcbgenomics.biomedcentral.com/articles/10.1186/s12864-022-08936-4>).

[These last 2 paragraphs also came up in the response to reviewer 1, we have copied here for ease of reading.]

3. The role of the priors on π and ω need to be described in a more quantitative manner in the main manuscript. Based on the data presented in SM, it appears that there is little information in the data to infer the distributions of these two parameters and therefore they are heavily impacted by the choice in prior. Fortunately, it looks like the impact on final serial interval estimates may be more robust, but this is only described in vague terms.

We have added a new supplementary analysis investigating the role of the π and w priors, using the Kullback-Leibler divergence of posterior->prior and posterior->likelihood to assess the relative influence on the posterior of likelihood & the prior (*Figure S7*). We confirm our previous findings that the priors are heavily influential on our estimates of π and w , now quantified to contributing around 75% of our ‘relative posterior influence’ (as defined with the KL divergence), but find that the prior is less influential in larger clusters.

We have added discussion of this in the main manuscript, alongside more interpretation of the previous (and updated) supplementary results [this also came up in the response to reviewer 1, we copy here for ease of reading]:

“A sensitivity analysis to these assumed prior distributions, included in the Supplemental Materials Section S2, reinforces that π and w are influenced by changes to their prior,

but our serial interval estimates are robust to moderate changes in these sampling rate priors (on the order of a 20% change to the prior mean or 50% to the prior standard deviation). We further quantify the influence of the priors on π and w by calculating the Kullback-Leibler (KL) divergence between the posterior and prior (the likelihood information) and between the posterior and likelihood (the prior information) [22] for all wave 1 clusters, shown in Figure S7. We again find that the priors are influential on our estimates of π and w (though this approach is not able to assess the onward impact to estimating the serial interval), with around 75% of the posterior influence coming from the prior and 25% from the likelihood, under our KL statistic. The likelihood is more influential in clusters with a larger number of identified cases (A2, A7), suggesting that with higher genomic surveillance, more data and therefore larger clusters, our estimates would be less reliant upon a predetermined prior sampling distribution.” line 213

And noted briefly again in the Discussion.

“However, the sensitivity analysis of our prior assumptions revealed that our results are robust to moderate deviations from the sampling rate priors (approximately, mean $\pm 20\%$, standard deviation $\pm 50\%$). Nonetheless, the sampling-related prior distributions for π and w should be chosen carefully, using knowledge of surveillance and sequencing in the population of interest” line 333

Additional concerns:

1. The beginning of the introduction is not very consistent with the focus of the paper. For example, all of the benefits of WGS data for outbreak investigation are focused on bacterial outbreaks (e.g., AMR tracking), while the manuscript is focused on viral outbreaks.

We have edited the first paragraph of the introduction, particularly to include more of a focus on viral pathogens:

“Whole genome sequence (WGS) data is rapidly becoming a fundamental tool in public health laboratories (PHL) around the world [1–3]. WGS data carry enormous benefits for outbreak investigations: identifying transmission events that were not detected during epidemiological study [4] and revealing the impact of border control measures [5], especially where data are shared across jurisdictional boundaries [3]. However, the information content of genomic data alone can be limited, as experienced during the SARS-CoV-2 pandemic [6, 7]. Often, genomic data are combined with epidemiological data in an ad hoc fashion by plotting epidemiological data on the tips of phylogenetic trees (derived from genomic data). This does not lend itself readily to desired PHL reproducibility and repeatability standards. On the other hand, when genomic surveillance data are systematically linked to epidemiological and clinical information, genomic epidemiological investigations can better inform public health action through a contextual understanding of population demographics, immunisation, clinical impacts, spatial transmission patterns, and more [8, 9].” line 23

2. Line 168: What is meant by “wild-type SARS-CoV-2”? Non variant of concern?

Yes, we have added “(non Variant of Concern [VOC])” here for clarity line 206

3. Are there also priors on the mean and standard deviation of serial interval, which could be displayed on the figures (e.g., Fig. 4)?

No, we do not apply priors for the serial interval, only for π and w which may be more reasonably be informed by prior knowledge of population sampling rate. We have added

mention of this to the text,

"We do not use priors for μ or σ " line 417

4. Figure 7: It is very difficult to discern any patterns with the way the data are currently displayed. I would suggest testing alternative visualization strategies. For example, sorting clusters by "Type" rather than estimate.

We have reordered Figure 7 (now *Figure 8*) by Type (and by mean serial interval within that).

5. Line 235: Clarify the meaning of "within one population." How is population defined?
After edits, this sentence no longer exists.

6. Line 265: Please quantify "moderate deviations."

We have added this,

"our serial interval estimates are robust to moderate changes in these sampling rate priors (on the order of a 20% change to the prior mean or 50% to the prior standard deviation)" line 215

and again in the Discussion

"...moderate deviations from the sampling rate priors (approximately, mean \pm 20%, standard deviation \pm 50%)" line 334

7. Line 288: "not limited to genomic analyses." This aspect of the paper is not developed in enough detail.

We expanded this section of the discussion.

"...As indicated by the secondary contact-based analysis, the estimation model presented here, which is novel in and of itself, is not limited to situations in which genomic data are used to identify potential pairs. If contact data is available, it can be used to build or inform the collection of feasible transmission networks, and thereby estimate the serial interval distribution. Our estimation model extends the work of Vink et al. [10] by allowing for any number of unsampled intermediate cases and removing the focus on the cluster index case. In this work, we explored how either genomic or contact data alone can teach us about transmission, but in practice a combination of data sources may result in the best estimates" line 358

This concept is also now developed more fully in the main text rather than the supplement, by inclusion of *Figure 9* and its associated text (*line 263*).

8. Line 298: What is meant by "low transmission divergence"?

We have added a definition here alongside the reference to the paper which introduced this term,

"Our method behaves well in settings with low transmission divergence (low number of mutations separating transmission pairs) [31]..." line 374.

Reviewer #3 (Remarks to the Author):

This study estimates the serial interval for COVID-19 cases in Victoria, Australia during the first two waves of the pandemic. The dataset is large, has extensive information from contact tracing, and the study addresses a question of broad interest. The authors employ a novel, probabilistic method of reconstructing transmission chains within

outbreak clusters and use it to infer the mean and standard deviation of the serial interval for individual case clusters, and from that derive the distribution of serial intervals for the entire dataset.

Conceptually, the study has two components: reconstruction of the transmission network, and estimation of the serial interval, with the latter dependent on the former. Both parts are (as the authors note) difficult, the first because of undersampling and low genetic diversity at the scale of individual transmissions, the second because the estimated serial interval is highly dependent on the (unknown) degree of undersampling.

My overall conclusion is that the paper is well written and the methods used are promising, but that more should be done to validate the accuracy of those methods.

Thank you for your helpful comments and suggestions. We have added several new validation analyses to the manuscript, and responded to each point below.

Two validation methods are described in the paper; both could be made more rigorous:

1) In the first method, some clusters are re-analyzed using only contact tracing data. This exercise yields plausible values for the serial intervals, which is reassuring. It would be better and more persuasive, however, if a direct comparison were made between the estimates from the two methods for each cluster (when such comparison is possible). This would permit a quantitative assessment of their agreement. (Based on the supplemental figure, the agreement is likely to be good.)

This is difficult because the contact traced clusters do not have much overlap with the genomic clusters (and there are cases who were contact traced but not sequenced). This has now been shown in *Figure S8* in supplement *Section S5*. However, we have added a new comparison of the serial intervals for those 4 clusters that share at least 50% of cases, in *Figure 9*. We find good agreement between the serial intervals estimated with genomic data and with contact data. Our estimates of π and w are higher in the contact data analysis, which is sensible given that a higher proportion of cases within our clusters were contact traced than successfully sequenced. This is discussed in the main text [this also came up in the response to reviewer 2, we copy here for ease of reading]:

“Although direct comparison is difficult due to the fact that the genomic and contact-based clusters do not entirely overlap, the estimated serial intervals are similar under both methods. This is especially true in those pairs of clusters which are most similar under the 2 clustering methods, sharing at least 50% of the same cases; presented here in Figure 9. We see good agreement among estimates for the serial interval mean and standard deviation. The cluster with most significant disagreement (C13/A2) was associated with several instances of international travel from different continents, leading to local transmission (cluster 70 in [6]). We estimate that the contact-defined clusters have a higher sampling proportion and lower proportion coprimary than their corresponding sequence-defined cluster. This is logical, given that we used larger prior means for π and w , as not all contact-traced cases were successfully genomically sequenced” line 265

2) The second validation method involves varying the two priors (on the non-coprimary transmission fraction and on the sampling fraction). When this is done, the posteriors for those parameters, but not for the serial interval, change substantially, indicating that the data provide little information about the two parameters but a lot of information about the serial interval. This is a surprising result, since the degree of undersampling should be directly reflected in the estimated serial interval. I believe the problem here is that both priors are changed at the same time and in the same direction (assuming I understand the description), leading to at least partial cancellation in the estimate of the serial interval. For example, reducing π from .5 to .4 increases the mean of $(m+1)$ from 2.0 to 2.5, leading to a 20% reduction in the estimate of μ ; reducing w at the same time increases the fraction assigned to coprimary transmission by 20%, leading to an increase in estimated μ . To really test dependence on the priors, they should be varied independently.

We have updated the sensitivity analysis to vary each prior distribution independently, now shown in *Figures S4 and S5*. We still find that the posteriors for π/w change substantially, but the serial interval estimates do not (although, they are more varied than previously, in the expected direction).

This is discussed in the main text,

“A sensitivity analysis to these assumed prior distributions, included in the Supplemental Materials Section S2, reinforces that π and w are influenced by changes to their prior, but our serial interval estimates are robust to moderate changes in these sampling rate priors (on the order of a 20% change to the prior mean or 50% to the prior standard deviation)” line 213.

Since this study introduces a novel algorithm for reconstructing transmission networks, the accuracy of that algorithm should also be validated directly. Given the extensive contact tracing data available for these samples, a direct comparison between the assigned probabilities of putative infector/infected pairs and the rate of validation by contact tracing could be done. Since there already exist many algorithms for this kind of reconstruction, it would also be appropriate to provide some text placing this novel approach within the framework of existing methods.

We have added such a comparison for the wave 1 data, as the wave 2 contact data is much more patchy. This is shown in *Figure S12*. We compared the assigned [infector, infectee] probabilities from the genomic analysis to the identified contact pairs and shared exposure sites from contact tracing data, for all infectees with at least 1 direct contact (40 infectees total). We find that infectors with a known contact link to the infectee are more likely to be selected in the transmission tree building, as they have higher assigned probabilities.

This is now discussed in the supplement:

“In Figure S12, we compare the probabilities with which each plausible infector is chosen to be the ancestor of each infectee in the genomic analysis (as defined in Section 5), against the contact tracing data, where available. Each infectee with at least one direct traced contact is shown as a panel, and all of their plausible infectors are shown with their assigned probability from the genomic analysis (y-axis) and contact type from the contact tracing data

(colour). For most infectees, there are many possible infectors, the majority of whom do not have any contact-tracing links. However, note that lack of an identified contact link does not necessarily correspond to lack of a link in reality. In many cases, we see that infectors with known contact to the infectee (shown in red) are assigned a higher probability, relative to the other potential infectors. The same is true for shared exposure sites to some extent, though less reliably. For example, out of 51 infectors with identified contact links to their plausible infectee, 40 are among the highest ranked infectors. For infector-infectee pairs with shared exposure site, 70/133 plausible infectors are among the highest ranked” Section S5

We have also added text to the introduction which places our approach in the context of existing outbreak reconstruction algorithms,
“While there exist several algorithms for outbreak reconstruction from genomic data in the context of sufficient genetic variation to construct well-resolved pairs or phylogenetic trees, for example the outbreaker, TransPhylo, SCOTTI and Beastlier platforms [16–19], these have been focused on densely-sampled outbreak settings where inference of who infected whom is the primary aim. Our approach is targeted at broader settings with lower levels of sampling and of genetic variation, where there may not be sufficient information in the data to reconstruct transmission pathways with high confidence. We therefore use a fast and simple model for pair reconstruction, followed by a statistical model that averages over the uncertainty in who infected whom to estimate the serial interval distribution. To the best of our knowledge, no existing outbreak reconstructive models have considered estimation of serial intervals in this context” line 70

And expanded where this is mentioned in the Discussion:

“It may be possible to do this by incorporating existing methodologies for sampling of transmission trees, such as the outbreaker or TransPhylo platforms [16, 17], but these are not well positioned to estimate serial intervals as the underlying transmission models do not consider the time of symptom onset” line 321.

Other issues:

Were there any asymptomatic cases in the dataset? How would these be handled, since the serial interval only applies to symptomatic cases?

This is definitely a challenge. As you noted, there is no obvious definition of the serial interval for asymptomatic cases. As a result, we filtered cases with missing symptom onset time (either arising from being asymptomatic or from data collection error) from the dataset [detailed in Methods e.g. *line 444, 467*]. Our serial intervals can then be seen as representative of infection to symptomatic cases. Where an infector was asymptomatic, they would appear as an unsampled intermediate case, and this in part motivated our prior for parameter π .

Asymptomatic cases have been found to transmit at lower but still significant levels compared to symptomatic cases (e.g. <https://www.sciencedirect.com/science/article/pii/S2666776221000338>), but the literature on this often combines true asymptomatic and presymptomatic (specifically, asymptomatic at the time of transmission but later symptomatic) cases, making it difficult to quantify the full extent of this. Presymptomatic transmitters would be included in our analysis.

We have added text on this to the Discussion,

“Another broad challenge in estimating serial intervals is how to incorporate asymptomatic cases, as these individuals naturally will not have a time of symptom onset. In this analysis, we removed all cases with no symptom onset time, and so our serial intervals can be thought of as representing transmission to symptomatic cases, with potentially-asymptomatic unsampled intermediates. Although asymptomatic cases have been identified as transmitters of SARS-CoV-2 [26] albeit at a reduced rate, many studies have not differentiated between true asymptomatic cases and cases who were asymptomatic at the time of transmission but later developed symptoms (presymptomatic, these individuals would be included in our analysis)” line 337

While any model requires arbitrary choices, some motivation could be provided for some of the choice. For example, the time window (35 days) allowed for sample pairs to be part of a cluster corresponds to ~7 viral generations, while the window in genetic distance (1.1 mutations/genome) is ~2 generations. Why the discrepancy? Similarly, the model incorporates 1) a single line of descent with any number of intermediate infections, and 2) co-primary transmission (pairs of samples infected from the same case), but does not allow for coprimary transmission with intermediates. Why the omission -- tractability? What kind of effect might the omission have?

Absolutely. We have added more motivation to the manuscript with respect to the first part of this comment [this also came up in the response to reviewer 1, we copy here for ease of reading]

“We apply a much more stringent genomic distance criterion (corresponding to approximately 2 generations of infection) than onset distance criterion (corresponding to approximately 7 generations), to allow the genomic data to be relatively more informative than the time-based data, and to minimise input bias towards short serial intervals” line 497.

For the second part, yes this is largely a combination of tractability and practicality. Whilst in theory it should be possible to consider coprimary transmission with intermediates (the folded gamma difference distribution would become a folded compound geometric gamma difference distribution), the calculations become significantly more cumbersome and at the same time the chance that any sampled transmission pairs would have a true history matching this pattern whilst also meeting the distance criteria discussed in the first part of the comment becomes low. We have added this to the discussion however, as it is certainly an extension to the method that may be helpful in future applications where the level of population sampling is lower,

“Lastly, the coprimary transmission model could be extended to include further unsampled intermediate cases, matching the non-coprimary model. This was not a priority in this work, as the high level of sampling makes such scenarios increasingly unlikely, but could be impactful in settings with a larger proportion of unsampled cases. If such settings did occur in the data studied here in reality, the impact would be that our serial intervals are overestimated” line 324.

Trivia:

I. 96: Should read 'inversely proportional'.

I. 400: Presumably this should say 'result in higher sampling probability'.
eqn. 1: The last term seems to be missing a μ .
Thank you, we have corrected these three errors.

Steve Schaffner

REVIEWER COMMENTS

Reviewer #2 (Remarks to the Author):

This is my second time reviewing "Genomic epidemiology offers high resolution estimates of serial intervals for COVID-19" by Stockdale et al. Overall, my impression is that the authors carefully considered the comments and questions from all three reviewers and made substantial changes that have improved the manuscript. I also feel that the methodology presented in the paper will be valuable for the community even though it has some important limitations. However, I'm not convinced that their application of the method to SARS-CoV-2 genomes has led to any significant advances in our understanding of this virus.

Prior to publication, there are still several things I would like to see the authors address:

1. The analysis of simulated data was a nice addition, but it has been used in a very limited capacity to explore the impact of incomplete case sampling. There are two other aspects that I would have liked to see explored with simulated data. First, it would have been nice to see variability in the serial interval across clusters, and how well the methodology was able to pick up on that variability. Second, I would have liked to see the influence of priors on π and ω explored more quantitatively using the simulated datasets. Using the SARS-CoV-2 datasets, the authors show that the priors used for π and ω have a large influence on the posterior distributions for these parameters (and some influence on other parameters). Given that they know the values of these for the simulated datasets, they could have quantified how improper priors would influence estimates of serial interval (and effective reproduction #).
2. Authors should indicate why the number of sampled transmission networks was 10x higher for the simulations (1000) compared to the SARS-CoV-2 datasets (100).
3. The meaning of the following is vague: "The mean of the priors is set to be within 10% of included proportion p , ..." Was the mean of the prior set at exactly the actual rate with 10% standard deviation?
4. To contextualize cluster-specific serial interval estimates, the authors calculate R_t values for different clusters over time. Given that they both utilize virus genomes to help estimate epidemiological parameters, I would like the authors to include some discussion of the differences/similarities between their method of estimating R_t and the method of Tanja Stadler et al., which utilizes birth-death models in BEAST.
5. In general, the figure legends are bare bones and could all benefit from more detailed descriptions of what is being shown.
6. I'm a bit confused by the meaning of the "primary" 10 clusters for wave 2. Is the only distinction that the results for these were shown individually, while the others were only shown in aggregate? This should be clarified.
7. The authors show that the larger the cluster, the less important the priors on model parameters. This left me wondering whether creating small clusters is a necessary first step, rather than simply analyzing the full dataset as a single cluster. Is this simply not feasible computationally? Could this be a future direction?
8. The authors should include some type of statistical test to accompany their discussion of the trends observed in Figure 8.

Reviewer #3 (Remarks to the Author):

The authors have comprehensively addressed my comments and I have no further issues to raise.

Response to reviewer: NCOMMS-22-11439A

Thank you to the reviewer for their feedback. We have responded to all points below, and added new analyses to the manuscript.

Where changes have been made in the manuscript, line references are provided in the response and refer to the resubmitted version.

Reviewer #2 (Remarks to the Author):

This is my second time reviewing “Genomic epidemiology offers high resolution estimates of serial intervals for COVID-19” by Stockdale et al. Overall, my impression is that the authors carefully considered the comments and questions from all three reviewers and made substantial changes that have improved the manuscript. I also feel that the methodology presented in the paper will be valuable for the community even though it has some important limitations. However, I’m not convinced that their application of the method to SARS-CoV-2 genomes has led to any significant advances in our understanding of this virus.

Prior to publication, there are still several things I would like to see the authors address:

1. The analysis of simulated data was a nice addition, but it has been used in a very limited capacity to explore the impact of incomplete case sampling. There are two other aspects that I would have liked to see explored with simulated data. First, it would have been nice to see variability in the serial interval across clusters, and how well the methodology was able to pick up on that variability. Second, I would have liked to see the influence of priors on π and ω explored more quantitatively using the simulated datasets. Using the SARS-CoV-2 datasets, the authors show that the priors used for π and ω have a large influence on the posterior distributions for these parameters (and some influence on other parameters). Given that they know the values of these for the simulated datasets, they could have quantified how improper priors would influence estimates of serial interval (and effective reproduction #).

We have added a new simulation study in which we explore, as suggested, the performance of our method under clusters with varying serial interval. We simulated a set of outbreaks with mean (1.2, 2, 3, 4, 5, 6) days [our simulator requires serial interval mean >1 day] and fixed standard deviation 1. We then estimated the serial interval (and π , w) for each simulated outbreak. In these simulations, we downsampled to 50% of true cases in the outbreaks and again used a prior on π and w with approximately correct mean but moderate uncertainty (SD 0.1) - so as to reflect a similar scenario to that in the real COVID-19 data.

We repeated our test of a significant difference between clusters (now, a cluster is a set of simulated outbreaks with the same serial interval distribution), and present these results in

the supplement. See supplement pages 18-19 and Figure S9 for the full details, but in short our findings are broadly,

- The method is able to distinguish the means
- Performance is not strong however when the true mean serial interval is only 1 day - the SI variance is artificially small because of our assumption that the SI must be positive
- We removed the mean 1.2 day serial interval from the test of significantly different means, and still were able to reject the null hypothesis of identical means. (In other words, the test was not simply detecting only the erroneous result for the smallest SI).

We now comment on point 2 in the Discussion

In our supplementary simulation study, we further found this assumption [positive serial intervals only] limits the method's ability to infer serial intervals with mean 1 day. However, this is infeasibly short for many diseases including COVID-19, see e.g. the published estimates in Table S2. Line 324

It is not the case that in our simulated data we know the true posterior distributions of π and w , because this is dependent on which cases we happen to sample and so will vary between different sampled outbreaks/networks. Although we could avoid this somewhat by simulating directly from the mixture model, this loses the connection to a simulated outbreak and its intricacies e.g. depletion of susceptibles.

However, we do know that π and w are related to the sampling proportion p , as in our previous simulation study, - if we believe the proportion of sampled and sequenced cases is lower, we expect lower π (more missing intermediates) and lower w (more coprimary pairs). So, we added a second experiment in which we simulate outbreaks with fixed serial interval and sampling proportion $p = 0.5$, but vary the prior we use. As our baseline prior on both π and w , we used a Beta(mean 0.5, SD 0.1) e.g. mean informed by the sampling rate, with some uncertainty. This is similar to our COVID-19 approach. We explored prior scenarios where we 'incorrectly' under- or over-estimated the mean of π/w , and where we used higher or lower prior variance.

The full details and results of this are presented in the supplement, alongside our previous analysis of sensitivity to the prior for the COVID-19 analysis. We found that the serial interval estimates were again fairly robust to use of incorrect priors, although our posterior estimates of π/w change. We concluded that underestimating π/w has the most negative impact on the mean serial interval estimates, although overall the relationships here are complicated by π/w not being equivalent to the sampling proportion p . When we enforced π & $w = 0.5$ with a less variable prior, we ended up overestimating the serial interval variance where fewer pairs were identified as having missing intermediates/coprimary infectors.

We added a comment about this in the Discussion

Nonetheless, the sampling-related prior distributions for π and w should be chosen carefully, using knowledge of surveillance and sequencing in the population of interest. Due to the potentially complex relationship between the proportion of cases sequenced and the parameters π and w (also affected e.g. by population structure), our simulation study exploring prior choice in section S2 suggested that diffuse priors may be preferable. Line 346

2. Authors should indicate why the number of sampled transmission networks was 10x higher for the simulations (1000) compared to the SARS-CoV-2 datasets (100).

This was in part a reflection of the larger population size in our simulations - we wanted to ensure we captured the (potentially larger) space of plausible networks. We have now explained this in Materials and Methods.

We sample a larger number of transmission networks (1000) than in the Victorian analysis (100), to reflect the larger population size, which may lead to a larger space of plausible networks. Line 427

3. The meaning of the following is vague: “The mean of the priors is set to be within 10% of included proportion p , ...” Was the mean of the prior set at exactly the actual rate with 10% standard deviation?

Apologies, we have rephrased this. We do not set the prior means exactly at the proportions p , but within 10% either side (varied across experiments) to represent solid but imperfect knowledge of the sampling rate. The prior standard deviation is always 0.1.

We use beta distributed prior distributions on p_i and w that represent moderate knowledge of the sampling rate, as this is the situation in which our method is intended to be most suitable. That is, the prior mean is set between 90%-110% of included proportion p (varied across experiments), with prior standard deviation set to 0.1. We note that p is not exactly equivalent to either p_i or w , the relationship will depend on the structure of the sampled network. We do not use priors for μ or σ . Line 431

4. To contextualize cluster-specific serial interval estimates, the authors calculate R_t values for different clusters over time. Given that they both utilize virus genomes to help estimate epidemiological parameters, I would like the authors to include some discussion of the differences/similarities between their method of estimating R_t and the method of Tanja Stadler et al., which utilizes birth-death models in BEAST.

We have added comments on this in the Discussion,

Although conceptually and methodologically quite different, this approach shares goals with the work of Stadler et al, 2012 [27] who estimate R_t from viral sequence data using a birth-death skyline model, via estimating time-varying transmission, recovery and sampling rates with phylodynamics in BEAST2 [28]. Unlike our approach, their focus is on reconstruction of the phylogeny without consideration of symptom onset times or the serial interval. Nonetheless an interesting extension could be to compare estimates of R_t from both methods. Line 311

5. In general, the figure legends are bare bones and could all benefit from more detailed descriptions of what is being shown.

We have added more detailed descriptions throughout

6. I'm a bit confused by the meaning of the "primary" 10 clusters for wave 2. Is the only distinction that the results for these were shown individually, while the others were only shown in aggregate? This should be clarified.

Essentially, yes. This is described in the text at line 163

There are a total of 94 wave 2 clusters, although for our main analysis we focus on 10 primary exposure site clusters comprising the largest 2 clusters associated with each of: aged care facilities, healthcare facilities, housing, schools, and meat packing/meat processing plants.

7. The authors show that the larger the cluster, the less important the priors on model parameters. This left me wondering whether creating small clusters is a necessary first step, rather than simply analyzing the full dataset as a single cluster. Is this simply not feasible computationally? Could this be a future direction?

One certainly could analyse the full dataset as a single cluster, although as mentioned it may be computationally infeasible depending on the dataset (for our data, it would be feasible for an entire wave so long as the pair criteria were reasonably stringent). The suitability of this would depend on to what extent we believe there was transmission between our clusters (potentially informed by epi data), if we are to allow between-cluster transmission pairs. We would of course lose the ability to compare the serial interval between clusters, which was one of our main aims here.

Our 'All'/pooled clusters analysis aims to be a middle ground between these 2 ideas: creating the transmission network by cluster (and hence restricting ourselves to transmission pairs within-cluster only) but then pooling all networks within each wave to estimate a single serial interval distribution.

We have noted this in the Discussion

... it could suggest clusters to focus resources upon, for example those which suggest a lower sampling rate, more rapid transmission, or substantially different or uncertain serial intervals. Our method is not restricted to use on small clusters: one could estimate the population level serial interval, although this would be more computationally demanding. Although reconstruction... Line 366

8. The authors should include some type of statistical test to accompany their discussion of the trends observed in Figure 8.

This is included in the supplemental materials, section S3 and Figure S8. We find that packing and meat processing plants have significantly shorter serial intervals than the other exposure category types. We have expanded on the description of this in the main text, *In the supplementary analysis, our statistical test reveals that packing and meat processing plants have statistically significant shorter serial interval than the other categories (Supplemental section S3 and Figure S8). Line 252*

REVIEWERS' COMMENTS

Reviewer #2 (Remarks to the Author):

I am satisfied with the changes the authors have made to the manuscript.